# METTL3 promotes homologous recombination repair and modulates chemotherapeutic response in breast cancer by regulating the EGF/RAD51 axis

Enjie Li[†], Mingyue Xia[†], Yu Du[†], Kaili Long, Feng Ji, Feiyan Pan, Lingfeng He, Zhigang Hu*, Zhigang Guo*

Jiangsu Key Laboratory for Molecular and Medical Biotechnology, College of Life Sciences, Nanjing Normal University, Nanjing, China

**Abstract** Methyltransferase-like 3 (METTL3) and N[6]-methyladenosine (m[6]A) are involved in many types of biological and pathological processes, including DNA repair. However, the function and mechanism of METTL3 in DNA repair and chemotherapeutic response remain largely unknown. In present study, we identified that METTL3 participates in the regulation of homologous recombination repair (HR), which further influences chemotherapeutic response in both MCF-7 and MDA-MB-231 breast cancer (BC) cells. Knockdown of METTL3 sensitized these BC cells to Adriamycin (ADR; also named as doxorubicin) treatment and increased accumulation of DNA damage. Mechanically, we demonstrated that inhibition of METTL3 impaired HR efficiency and increased ADR-induced DNA damage by regulating m6A modification of EGF/RAD51 axis. METTL3 promoted EGF expression through m6A modification, which further upregulated RAD51 expression, resulting in enhanced HR activity. We further demonstrated that the m6A 'reader,' YTHDC1, bound to the m6A modified EGF transcript and promoted EGF synthesis, which enhanced HR and cell survival during ADR treatment in BC. Our findings reveal a pivotal mechanism of METTL3-mediated HR and chemotherapeutic drug response, which may contribute to cancer therapy.

*For correspondence:
huzg_2000@126.com (ZH);
guo@njnu.edu.cn (ZG)

[†]These authors contributed equally to this work

## Editor's evaluation

This paper identified a new mechanism by which N6-methyltransferase METTL3 modulates DNA repair and cellular resistance to DNA damaging chemotherapeutics. The mechanism involves upregulation of epidermal growth factor expression, which in turn regulates expression of RAD51 recombinase resulting in enhanced DNA repair by homologous recombination.

## Introduction

m[6]A modification of RNA has been reported to participate in regulating numerous cellular processes in eukaryotes (*Wang et al., 2014*; *Wang et al., 2015*; *Xiang et al., 2017*). METTL3 is a key member of the m[6]A methyltransferase complex, which also includes co-factors METTL14 and the Wilms tumor 1 associated protein (WTAP) (*Liu et al., 2014*; *Wang et al., 2015*). This RNA modification can be recognized by a set of m[6]A-binding proteins, including YTH domain containing family protein (YTHDF1/2/3), YTH domain containing 1/2 (YTHDC1/2), and insulin like growth factor 2 mRNA binding protein (IGF2BP1/2/3), which serve as m[6]A 'readers' and mediate specific functions of m[6]A-modified RNA (*Deng et al., 2018*; *Wang et al., 2015*). Furthermore, fat mass and obesity-associated protein (FTO) and RNA demethylase, ALKBH5, work as m[6]A 'erasers' to remove m[6]A modifications from RNA (*Deng*

et al., 2018). Recent studies have indicated that the m⁶A modification and METTL3 play important roles in the progression and chemotherapy response of various cancers, including BC (*Cai et al., 2018*; *Deng et al., 2018*; *Pan et al., 2021*).

Chemotherapy is used in early-stage BC and locally advanced BC to provide an improved chance for breast-conserving surgery, reduce the risk of recurrence, and increase survival rates (*Fisusi and Akala, 2019*). The use of adjuvant chemotherapy remains an effective treatment for triple-negative BC and other types of invasive breast cancer (*Hennigs et al., 2016*). Several chemotherapeutic agents, including ADR, docetaxel, 5-fluorouracil, and cisplatin are used in combination chemotherapy for BC treatment (*Fisusi and Akala, 2019*). Since 1970s, ADR was considered as the most active chemotherapeutic agent for the treatment of BC, which was used in neoadjuvant chemotherapy or combination therapy (*Chan et al., 1999*; *Fisher et al., 1998*; *Paridaens et al., 2000*). Chemotherapeutic agents, such as ADR, induce apoptosis by causing DNA damage (*He et al., 2016*; *Lu et al., 2020*). Targeting the DNA damage response (DDR) may enhance the sensitization of cancer cells to chemotherapeutic drugs (*Li et al., 2021*; *Lu et al., 2020*). Moreover, elevated DNA repair activity contributes to the drug resistance of cancer cells treated with chemotherapy (*Li et al., 2021*; *Lu et al., 2020*).

Key proteins involved in DNA repair, such as BRCA1, RAD51, and RAD52 in HR, xeroderma pigmentosum group C (XPC) in nucleotide excision repair (NER), and flap endonuclease 1 (FEN1) in base excision repair (BER), influence the susceptibility of various cancers and may be suitable targets in cancer mono- and combination therapy (*Ali et al., 2017*; *Grundy et al., 2020*; *Hengel et al., 2016*; *Huang et al., 2016*; *Li et al., 2021*; *Lu et al., 2020*; *Malik et al., 2020*; *Miki et al., 1994*). In addition, mutations in several DDR proteins, such as BRCA1, BRCA2, PALB2, and RAD51 also contribute to hereditary breast and ovarian cancer (*Hengel et al., 2017*; *Li et al., 2021*; *Lok et al., 2013*). Mutations in these proteins incapacitates HR and results in synthetic lethality with inhibition of RAD52 or PARP1, suggesting a novel strategy for treating patients with BRCA1/2/PALB2-mutant tumors (*Bryant et al., 2005*; *Farmer et al., 2005*; *Lok et al., 2013*). However, targeting key DDR proteins to improve sensitization of cancer cells and circumvent cancer cell resistance remain significant challenges to achieving satisfactory therapeutic effects (*Li et al., 2021*; *Pan et al., 2021*). Therefore, exploring novel treatment strategies and mechanisms that affect DDR in cancer cells may contribute to improved treatment for these diseases.

Currently, METTL3 and m⁶A modification have been implicated in DDR (*Xiang et al., 2017*; *Yu et al., 2021b*; *Zhang et al., 2020*). However, the molecular mechanism of METTL3 and m⁶A modification in DDR, especially in the context of chemotherapeutic drug-induced DDR, remains unexplored. Here, we report that METTL3 is involved in the regulation of HR by regulating the EGF/RAD51 axis. Knockdown of METTL3 sensitizes both MCF-7 (estrogen receptor (ER)-positive BC cell) and MDA-MB-231 (MB-231, triple-negative BC cell) BC cells to ADR, impairs HR, and induces significant DNA damage. Moreover, YTHDC1 was identified as the reader that binds and protects m⁶A-modified EGF mRNA and regulates DNA repair and the response of BC cells to ADR. Overall, our findings provide insight into the function and mechanism of METTL3 in HR and the chemotherapeutic drug response in BC, and demonstrate the potential of targeting METTL3 as an antitumor treatment.

## Results

### METTL3 regulates chemotherapeutic response of BC cells

METTL3 has been reported to be involved in the progression of several types of cancers, including BC (*Deng et al., 2018*; *Wang et al., 2020a*). We wonder whether METTL3 regulates chemotherapeutic response of BC cells. First, we identified the elevated METTL3 and m⁶A levels in five types of BC cells, including MCF-7, MB-231, T47D, SKBR3, and BT474 cells (*Figure 1—figure supplement 1A, B*). Then, we investigated the role of METTL3 in regulating the sensitivity of both MCF-7 and MB-231 cells (ER-positive and triple-negative cells, respectively) to chemotherapeutic drugs with stable METTL3-OV or -KD cell lines (*Figure 1—figure supplement 1C*, D). Cell viability assays were performed using METTL3-KD and METTL3-OV MCF-7 stable cell lines treated with five first-line chemotherapeutic drugs including 5-FU, cisplatin (DDP), ADR, paclitaxel, and carboplatin. The results indicated that modification of METTL3 expression markedly attenuated the sensitivity of MCF-7 cells to ADR compared with the other drugs (*Figure 1A* and *Figure 1—figure supplement 1E-I*; *Andreetta et al., 2010*). Cell viability assays using METTL3-OV and -KD MB-231 stable cells verified the effect of METTL3 on ADR

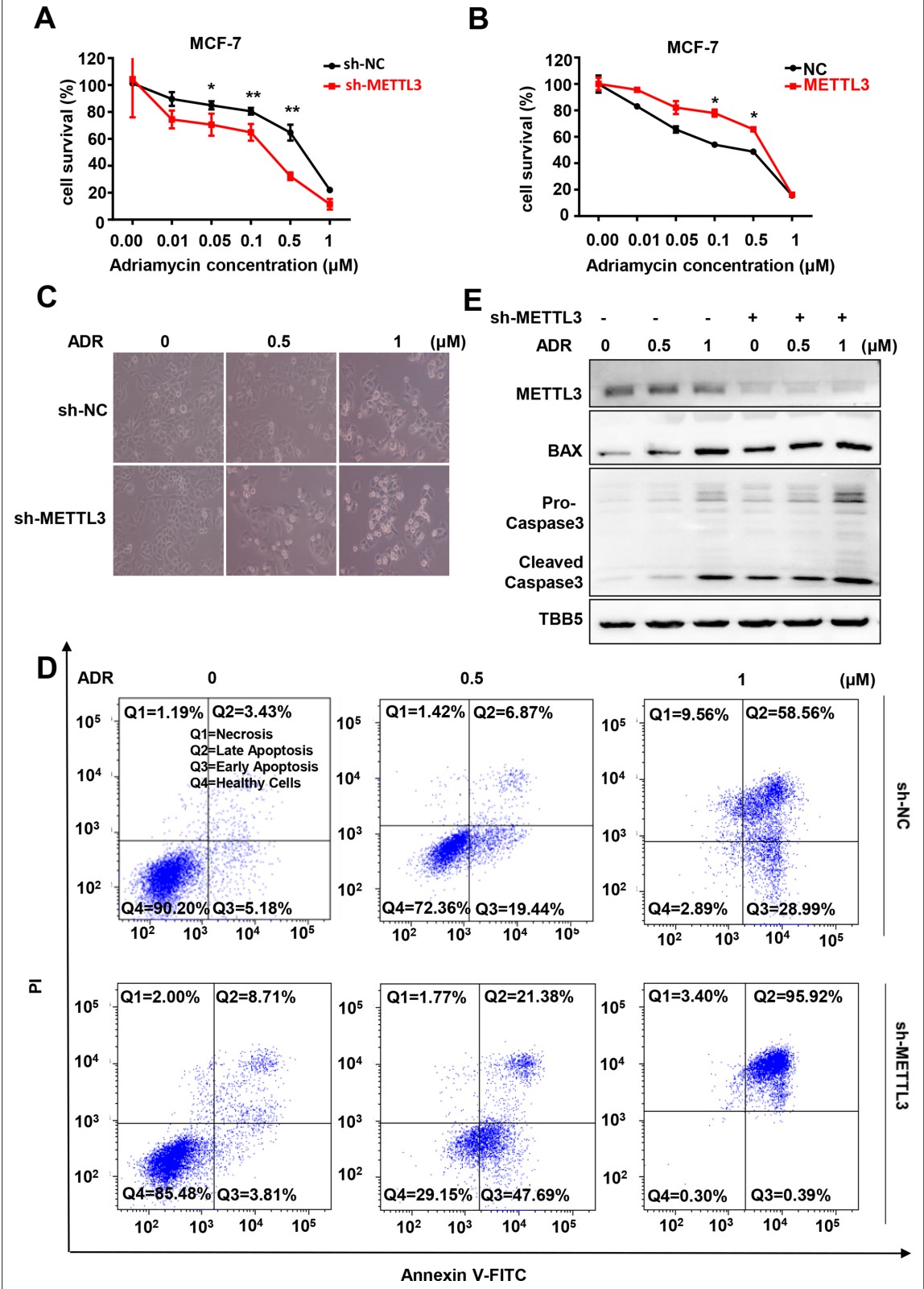

**Figure 1.** Knockdown of Methyltransferase-like 3 (METTL3) sensitizes both MCF-7 and MB-231 cells to Adriamycin (ADR). (**A, B**) MTT assays were performed to determine the effect of METTL3 on ADR cytotoxicity in MCF-7 or MDA-MB231 cells. Data are expressed as the mean ± standard deviation (SD), n=3 per group. (**C**) Morphological analysis of MCF-7 with different drug treatments. (**D**) Annexin V/PI staining and flow cytometry assay of control or

*Figure 1 continued on next page*

*Figure 1 continued*

METTL3-KD MCF-7 cells with different drug treatments. (**E**) Western blot (WB) analysis of METTL3, BAX, and Caspase 3 in control or METTL3-KD MCF-7 cells treated with various concentrations of ADR. All statistical data are presented as the mean ± SD. * p<0.05; ** p<0.01; *** p<0.001 (Student's *t*-test).

The online version of this article includes the following source data and figure supplement(s) for figure 1:

**Source data 1.** Source data for *Figure 1E*.

**Figure supplement 1.** Knockdown of Methyltransferase-like 3 (METTL3) sensitizes MCF-7 and MB231 cells to Adriamycin (ADR).

**Figure supplement 1—source data 1.** Source data for *Figure 1—figure supplement 1A*.

**Figure supplement 1—source data 2.** Source data for *Figure 1—figure supplement 1B*.

**Figure supplement 1—source data 3.** Source data for *Figure 1—figure supplement 1C*.

**Figure supplement 1—source data 4.** Source data for *Figure 1—figure supplement 1D*.

**Figure supplement 1—source data 5.** Source data for *Figure 1—figure supplement 1L*.

**Figure supplement 1—source data 6.** Source data for *Figure 1—figure supplement 1O*.

**Figure supplement 1—source data 7.** Source data for *Figure 1—figure supplement 1P*.

sensitivity (*Figure 1B* and *Figure 1—figure supplement 1E*). These results were further verified in a morphological analysis, which showed that silencing METTL3 enhanced chemotherapeutic drug sensitivity (*Figure 1C*), whereas overexpression of METTL3 decreased sensitivity in MCF-7 cells (*Figure 1—figure supplement 1J*). Furthermore, flow cytometry analysis demonstrated that treatment with ADR induced higher apoptosis rates in both METTL3-KD MCF-7 and MB231 cells compared with control cells (*Figure 1D* and *Figure 1—figure supplement 1K*). Accordingly, elevated pro-apoptotic Bax and caspase 3 were detected in METTL3-KD BC cells treated with ADR (*Figure 1E*). We then detected the effect of METTL3 on ADR-sensitivity using a human non-tumorigenic breast epithelial cell line MCF-10A. Our data showed that overexpression of METTL3 attenuated the sensitivity of MCF-10A cells to ADR (*Figure 1—figure supplement 1L-N*). As expected, silence of METTL3 reduced global m$^6$A levels in MCF-7 and MB-231 BC cells, whereas overexpression of METTL3 increased m$^6$A levels in these cells compared with control cells (*Figure 1—figure supplement 1O, P*).

## METTL3 promotes HR

ADR is normally described as a classic topoisomerase II poison that intercalates into DNA and forms DNA adducts, and subsequently induces DNA double strand breaks (DSBs) (*Swift et al., 2006*; *Yang et al., 2014*). We further addressed whether METTL3 was involved in DSB repair and affected ADR-induced DNA damage. To explore the role of METTL3 in the regulation of DNA repair, we treated METTL3-KD or –OV BC cells with ADR and subsequently released the cells into fresh medium lacking ADR and monitored the levels of γ-H2AX (an established marker of DNA damage) over time. Our data showed that knockdown of METTL3 maintained higher γ-H2AX levels in both MCF-7 and MB-231 cells compared with control cells (*Figure 2A and B*), whereas overexpression of METTL3 resulted in an earlier decline of γ-H2AX compared with control cells (*Figure 2—figure supplement 1A, B*). We further detected the effect of METTL3 on the regulation of DSB repair that was induced by etoposide (ETO; another inhibitor of topoisomerase II). Accordingly, our data showed that METTL3 promoted the repair of DSB induced by ETO in both MCF-7 and MB-231 cells (*Figure 2C and D*, *Figure 2—figure supplement 1C, D*). As showed in our data, the DNA damage was more rapid induced by etoposide than doxorubicin (*Figure 2A–D*), which was consistent with previous study in lung cancer cell lines (*Binaschi et al., 1990*). These results may due to a few differences among mechanisms of these two compounds poisoned DNA strands and cells, by which ADR induced cell death by trapping topoisomerase II, formation of ADR-DNA adducts, and generation of free radicals or ceramide production, whereas DNA damage secondary to topoisomerase II inhibition appears to be a major mechanism for etoposide-induced apoptosis (*Yang et al., 2014*; *Yang et al., 2001*). Consistently, an increased number of positive nuclei foci of γ-H2AX was detected in both METTL3-KD MCF-7 and MB-231 cells after ADR treatment and then released after 4 hr (*Figure 2E*), whereas experiments using METTL3-OV MCF-7 and MB-231 cells showed the opposite effects (*Figure 2—figure supplement 1E*). Since the phosphorylation and foci formation of γ-H2AX was the marker of both DNA damage and DNA replication stress, we also detected another DNA damage marker 53BP1 (tumor-suppressor p53-binding protein 1, a key regulator of DSB repair) foci (*Gagou et al., 2010*; *Wang et al., 2002*; *Ward and*

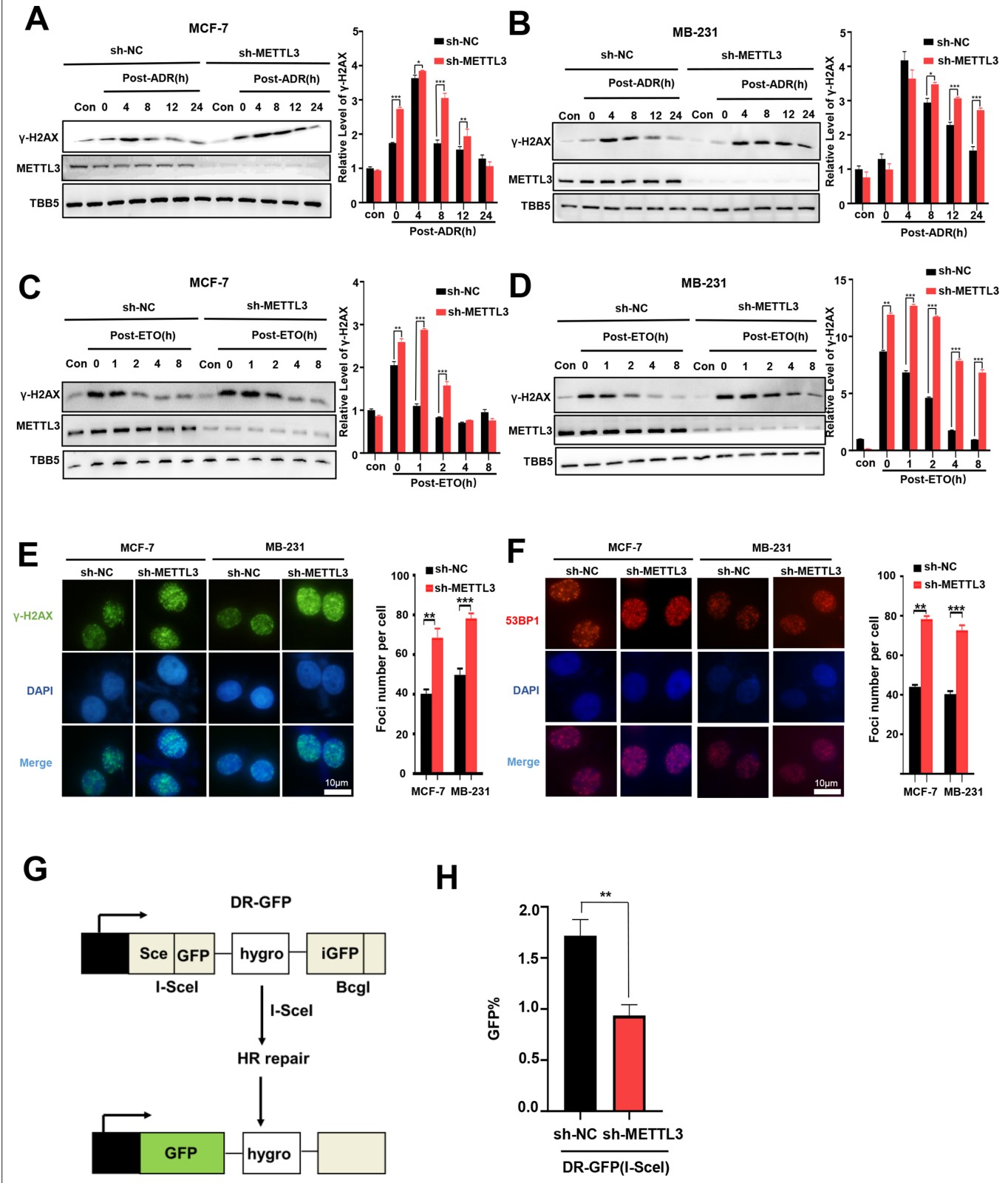

**Figure 2.** Knockdown of Methyltransferase-like 3 (METTL3) impairs homologous recombination repair (HR) efficacy. (**A, B**) Western blot (WB) assay to determine γ-H2AX levels in control and METTL3-KD MCF-7 cells (**A**) or MB-231 cells (**B**) with ADR (0.5 μM) treatment for 1 hr following different recovery times. (**C, D**) WB to determine γ-H2AX levels in control and METTL3-KD MCF-7 cells (**C**) or MB-231 cells (**D**) with ETO (10 μM) treatment for 1 hr following different recovery times. The Quantification of relative WB band are represented as the mean ± SD of three biological repeats. n=3 per group.

*Figure 2 continued on next page*

*Figure 2 continued*

(**E**) Immunofluorescence staining of γ-H2AX foci in different labeled cells with ADR treatment for 1 hr following 8 hr of recovery. (**F**) Immunofluorescence staining of 53BP1 foci in different cells treated same in (**E**). The quantification of average foci numbers per cells were showed in right panel, 50 cells were calculated in each group. (**G**) Schematic of GFP-based HR reporter system. (**H**) The GFP+ frequency of HR-mediated DSB repair in control and METTL3-KD U2OS cells (n=3 per group). ** p<0.01; *** p<0.001 (Student's *t*-test).

The online version of this article includes the following source data and figure supplement(s) for figure 2:

**Source data 1.** Source data for *Figure 2A*.

**Source data 2.** Source data for *Figure 2B*.

**Source data 3.** Source data for *Figure 2C*.

**Source data 4.** Source data for *Figure 2D*.

**Figure supplement 1.** Methyltransferase-like 3 (METTL3) enhances homologous recombination repair (HR) activity.

**Figure supplement 1—source data 1.** Source data for *Figure 1—figure supplement 1A*.

**Figure supplement 1—source data 2.** Source data for *Figure 2—figure supplement 1B*.

**Figure supplement 1—source data 3.** Source data for *Figure 2—figure supplement 1C*.

**Figure supplement 1—source data 4.** Source data for *Figure 2—figure supplement 1D*.

*Chen, 2001*). Similar results were obtained for the foci of 53BP1 in both MCF-7 and MB-231 cells (*Figure 2F* and *Figure 2—figure supplement 1F*). DSBs are primarily repaired by either HR or non-homologous end joining (NHEJ) (*Sonoda et al., 2006*). Using two well-characterized green fluorescent protein (GFP)-based HR and NHEJ reporter systems, we determined the effect of METTL3 on HR and NHEJ efficiency (*Mendez-Dorantes et al., 2020*; *Tsai et al., 2020*). The results showed that overexpression of METTL3 significantly enhanced HR-mediated DSB repair, whereas knockdown of METTL3 decreased efficiency of HR (*Figure 2G and H*; and *Figure 2—figure supplement 1G*). No effect of METTL3 was observed on NHEJ-mediated DSB repair (*Figure 2—figure supplement 1H, I*). This is consistent with previous reports using the HR and NHEJ luciferase reporter system based on crisper-cas9-induced DSBs (*Zhang et al., 2020*).

## EGF is the target of METTL3 and is regulated by m⁶A modification

A comprehensive assay combined with RNA-seq, MeRIP-qPCR, bioinformatics analysis, and literature retrieval were designed to explore the putative target(s) of METTL3-mediated m⁶A modification, which is involved in the regulation of both DNA repair and BC sensitivity to ADR (*Figure 3A*). A total of 98 genes showed significant changes (p<0.05; 42 up-regulated; and 56 down-regulated) in METTL3-overexpressing MCF-7 cells compared with control MCF-7 cells (*Figure 3B*). Among these genes, 52 were shown to be modified by m⁶A in the exonic, 5′UTR, or 3′UTR of mRNA region in the m⁶A-Atlas, a comprehensive knowledgebase for unraveling the m⁶A epitranscriptome (*Table 1*; *Tang et al., 2021*). Furthermore, literature retrieval identified 8 (*Table 2*) out of 52 genes that were reported to be involved in the regulation of DNA repair, among which EGF was highlighted because of its role in cancer progression, and DNA repair (*Myllynen et al., 2011*; *Wilson et al., 2009*; *Yacoub et al., 2003*). Next, we verified the expression of EGF regulated by METTL3. The mRNA levels of EGF increased in both METTL3-OV MCF-7 and MDA-MB231 cells compared with control cells (*Figure 3C and D*), whereas knockdown of METTL3 down-regulated EGF expression (*Figure 3—figure supplement 1A, B*). WB analysis of whole cell lysates further verified the up-regulation of EGF by METTL3 (*Figure 3E and F*, and *Figure 3—figure supplement 1C, D*). Moreover, secreted EGF in the culture supernatants were examined by ELISA. The results indicated increased EGF levels in the medium of METTL3-OV MCF-7 and MB-231 cells (*Figure 3G and H*), whereas down-regulated EGF was detected in the medium of METTL3-KD cells (*Figure 3—figure supplement 1E, F*). Overexpression of METTL3 also enhanced the expression of EGF in MCF-10A cells (*Figure 3—figure supplement 1G, H*). Furthermore, to validate the m⁶A modification in EGF mRNA, we performed a methylated RNA immunoprecipitation (meRIP)-qPCR assay using an m⁶A antibody followed by qPCR for the predicted region of the m⁶A sites. EGF mRNA exhibited the highest score by the sequence-based RNA adenosine methylation site predictor algorithm (*Zhou et al., 2016*). Using specific primers designed for the predicted m⁶A-harboring regions of EGF, the qPCR data showed that overexpression of METTL3 upregulated the m⁶A modification of EGF mRNA (*Figure 3I* and *Figure 3—figure supplement 1I*),

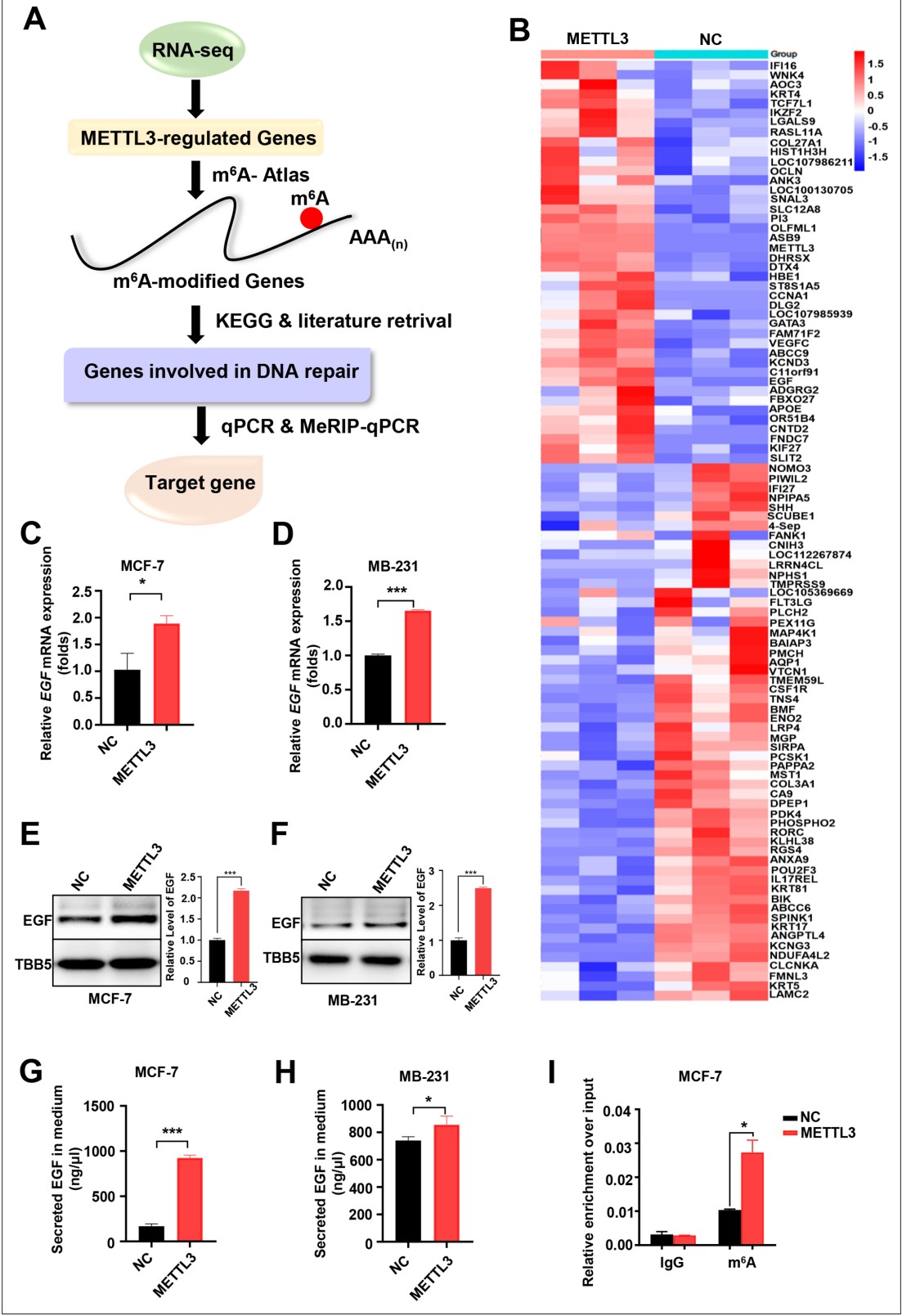

**Figure 3.** Screening and Identification of EGF as the targets of Methyltransferase-like 3 (METTL3) in breast cancer (BC). (**A**) Schematic of the screening progress of METTL3 targets in BC. (**B**) Heat map of RNA-seq to identify the genes regulated by METTL3 overexpression. The value presented Log$_2$ (fold change). (**C, D**) qRT-PCR was performed in METTL3-overexpressing MCF-7 (**C**) and MB-231 cells (**D**) to detect EGF expression. Data are expressed as the mean ± standard deviation (SD), n=3 per group. (**E, F**) Western blot (WB) analysis of EGF expression in METTL3-overexpressing MCF-7 (**E**) and MB-231

*Figure 3 continued on next page*

*Figure 3 continued*

cells (**F**). (**G, H**) ELISA assay measuring secreted EGF in the medium of METTL3-overexpressing MCF-7 (**G**) and MB-231 cells. (**H**) Data are expressed as the mean ± standard deviation (SD), n=3 per group. (**I**) MeRIP-qPCR analysis was used to assess the m$^6$A levels of EGF mRNA in METTL3-overexpressing MCF-7 cells. The enrichment of m$^6$A in each group was calculated by m$^6$A-IP/input and IgG-IP/input. Data are expressed as the mean ± standard deviation (SD), n=3 per group. * $p<0.05$; *** $p<0.001$ (Student's $t$-test).

The online version of this article includes the following source data and figure supplement(s) for figure 3:

**Source data 1.** Source data for *Figure 3E*.

**Source data 2.** Source data for *Figure 3F*.

**Figure supplement 1.** EGF is the targets of Methyltransferase-like 3 (METTL3) in MCF-7, MB-231, and MCF-10A cells.

**Figure supplement 1—source data 1.** Source data for *Figure 3—figure supplement 1C*.

**Figure supplement 1—source data 2.** Source data for *Figure 3—figure supplement 1D*.

**Figure supplement 1—source data 3.** Source data for *Figure 3—figure supplement 1G*.

whereas knockdown of METTL3 significantly attenuated its m$^6$A levels (***Figure 3—figure supplement 1J***). Our data indicated that EGF was the target of METTL3 and regulated by METTL3-mediated m$^6$A modification in both BC cell line and non-tumorigenic breast cell line.

## EGF regulates RAD51 expression and enhances HR activity

The EGF/EGFR signaling pathway has been reported to regulate DSB repair in lung cancer cells following X-irradiation by promoting both the NHEJ and HR pathways (***Kriegs et al., 2010***; ***Myllynen et al., 2011***). Thus, we wondered whether EGF/EGFR is involved in METTL3-mediated DSB repair in both MCF-7 and MB-231 cells treated with chemotherapeutic agents, such as ADR. First, we evaluated the effect of the EGF/EGFR signaling pathway on DSB repair in these BC cells. MCF-7 and MB-231 cells were treated with ADR or ETO, respectively, and then released for different times with EGF treatment. WB analysis showed that the γ-H2AX levels markedly decreased in both ADR- and ETO-treated cells following EGF treatment, indicating that EGF enhanced DSB repair (***Figure 4A–D***). Using a GFP-based HR reporter system, we detected the effect of the EGF/EGFR pathway on regulating HR activity. Our data showed that additional EGF enhanced HR activity in the reporter system (***Figure 4E***), whereas the EGFR inhibitors erlotinib and gefitinib could inhibited the HR activity with lower GFP-positive cells compared to vehicle treatment (***Figure 4—figure supplement 1A, B***). To explore the molecular mechanism of EGF-mediated HR, we determined whether EGF/EGFR regulated the expression of core genes involved in HR, including *BRCA1*, *BRCA2*, *CtIP*, and *RAD51*. Our data showed that both EGF and METTL3 exhibited a slight effect on the regulation of BRCA1, BRCA2, and CtIP expression (***Figure 4—figure supplement 1C, D***). In contrast, the expression of RAD51 was markedly regulated by the EGF/EGFR signaling pathway (***Figure 4F and G***, and ***Figure 4—figure supplement 1***).

**Table 1.** 52 genes were showed to be modified by m$^6$A in the exonic in this study.

| KCNG3 | DLG2 | SLC12A8 | PHOSPHO2 | AQP1 | ABCC6 |
|---|---|---|---|---|---|
| RGS4 | C11orf91 | GATA3 | RORC | SIRPA | COL3A1 |
| ASB9 | HIST1H3H | IFI16 | KRT17 | BMF | |
| ST8SIA5 | FBXO27 | RASL11A | LAMC2 | BIK | |
| METTL3 | SLIT2 | DTX4 | TMPRSS9 | SHH | |
| ABCC9 | AOC3 | ANK3 | PLCH2 | NOMO3 | |
| APOE | COL27A1 | KIF27 | BAIAP3 | NPIPA5 | |
| EGF | OCLN | ENO2 | FMNL3 | MST1 | |
| IKZF2 | PI3 | PCSK1 | LRP4 | PMCH | |
| DHRSX | VEGFC | TNS4 | PDK4 | CSF1R | |

**Table 2.** Eight genes were showed to be modified by m⁶A and be involved in DNA repair in this study.

| EGF | METTL3 | DLG2 | VEGFC |
|-----|--------|------|-------|
| GATA3 | KRT17 | IFI16 | MST1 |

## METTL3-modification of DNA repair is EGF/ RAD51 dependent

We next explored the effect of EGF/EGFR signaling on METTL3-mediated HR in MCF-7 and MB-231 cells. WB analysis showed that knockdown of METTL3 down-regulated RAD51 expression, which was recovered by EGF treatment in both MCF-7 and MB-231 cells (*Figure 5A and B*). In contrast, overexpression of METTL3 up-regulated RAD51 expression, which was repressed by the EGFR inhibitors, erlotinib and gefitinib (*Figure 5—figure supplement 1A, B* ). Both EGF and RAD51 levels were down-regulated in xenograft tissues derived from METTL3-KD MCF-7 cells (*Figure 5C*). Accordingly, EGF treatment impeded METTL3-KD-mediated repression of DNA repair in both MCF-7 and MB-231 cells (*Figure 5D and E*). The EGFR inhibitor, erlotinib, repressed DNA repair activities that were up-regulated by overexpression of METTL3 in both BC cells (*Figure 5—figure supplement 1C, D*). These data indicate that the EGF/EGFR signaling pathway regulates RAD51 expression and participates in METTL3-mediated HR.

We next investigated whether the effects of METTL3 on DNA repair were EGF/RAD51 dependent. We first confirmed the effect of RAD51 on HR activity in GFP-based HR reporter system as showed in *Figure 2G*. Our data showed that overexpression of RAD51 promoted HR activity, whereas knockdown of RAD51 repressed HR efficiency in the reporter system (*Figure 5—figure supplement 1E, F*), which were consistent with other current studies (*Asan et al., 2019*; *Ouyang et al., 2021*). Then, we knocked down of RAD51 in METTL3-OV MCF-7 cells. The METTL3-OV MCF-7 cells were transfected with shRAD51 (shRNA of RAD51) for 36 hr, then treated with ADR for 1 hr and released for 8 hr with or without gefitinib/erlotinib treatment. Overexpression of METTL3 resulted in elevated DNA repair activity (shown by γ-H2AX down-regulation) which was reversed by treatment with siRAD51 or gefitinib/erlotinib (*Figure 5F*, and *Figure 5—figure supplement 1G*). We further detected the γ-H2AX foci and RAD51 foci in different treated cells by immunofluorescence. We stained Cyclin A to represent S/G2 phase cells. Consistently, immunofluorescence analysis showed that the γ-H2AX foci were down-regulated by overexpression of METTL3, which were reversed by treatment with shRAD51 or gefitinib in both cyclin A positive cells and cyclin A negative cells (*Figure 5G*). The 53BP1 foci showed similar partners to γ-H2AX foci in these different treated cells (*Figure 5—figure supplement 1H*). The RAD51 foci were also augmented by overexpression of METTL3 and reversed with gefitinib treatment in both cyclin A positive cells and cyclin A negative cells (*Figure 5H*). Moreover, we detected RAD51 foci in cells with single inhibition of METTL3 by shRNA or single inhibition of EGF/EGFR by gefitinib or double treatments with shMETTL3 and gefitinib. We found that the RAD51 foci decreased to similar levels in cells in these three conditions compared to those in control cells, which indicated an epistatic effect of shMETTL3 and EGF/EGFR inhibition (*Figure 5—figure supplement 1I*). This epistatic effect was verified by cells survival assay (*Figure 5—figure supplement 1J*). Furthermore, we detected γ-H2AX levels in the ADR-treated MCF-7 and MB-231 cells with overexpression of METTL3 or EGF or RAD51. Our data showed that overexpression of METTL3/EGF/RAD51 all could alleviate DNA damage (shown by decreased γ-H2AX) in both MCF-7 and MB-231 cells with similar manner (*Figure 5—figure supplement 1K, L*). These results suggest that METTL3 augments HR in ADR-treated BC cells via the EGF/ RAD51 axis.

## YTHDC1 enhances the METTL3/m⁶A-regulated EGF/Rad51 axis

There are two major families of m⁶A 'readers' that play a specific role in controlling the fate of methylated mRNA including the YTH family and the IGF2BP family (*Deng et al., 2018*; *Wang et al., 2015*; *Xiao et al., 2016*). To elucidate the specific m⁶A readers of EGF mRNA and to determine the m⁶A-dependent mechanism of EGF regulation, we performed qPCR assays to screen EGF-related m⁶A readers. Interestingly, knockdown of YTHDC1, but not other members of the YTH family or the IGF2BP family, down-regulated both EGF and RAD51 in MCF-7 cells (*Figure 6A* and *Figure 6—figure supplement 1A-C*). Furthermore, knockdown of YTHDC1 reversed the METTL3-mediated up-regulation of EGF and RAD51 (*Figure 6B and C*). The potential binding motif of YTHDC1 in EGF mRNA is UGG(m⁶A) CU, which is the preferentially binding motif of YTHDC1 (*Xu et al., 2014*). Using RIP-qPCR, we found that the direct interaction between YTHDC1 and EGF transcripts was enhanced in METTL3-OV cells

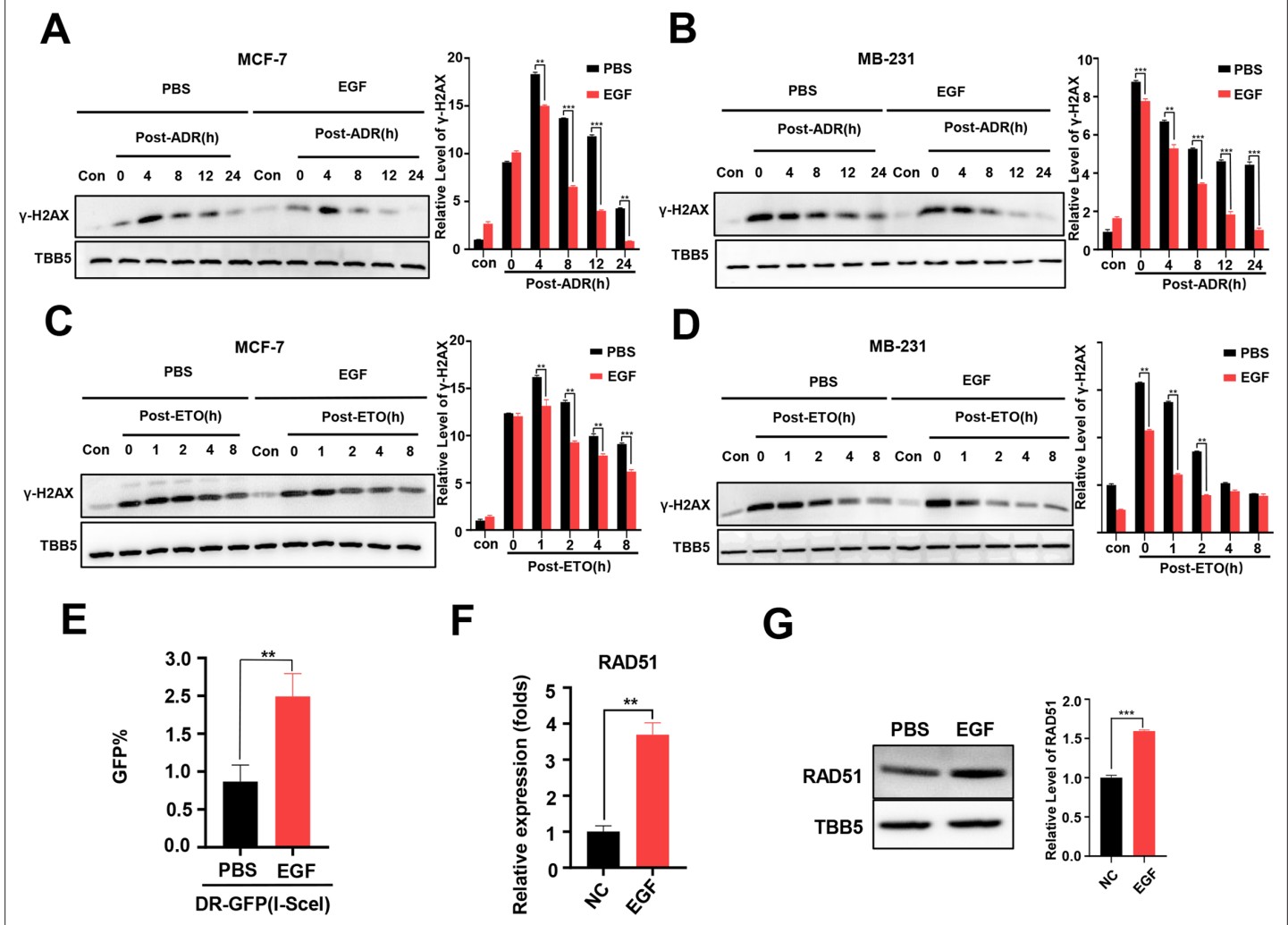

**Figure 4.** Additional EGF promotes RAD51 expression and enhances homologous recombination repair (HR) activity. (**A, B**) Western blot (WB) assay identified that EGF (10 ng/ml) enhanced DNA repair in MCF-7 (**A**) and MB-231 cells (**B**) treated with ADR (0.5 μM) (shown by lower γ-H2AX in EGF treated samples compare to PBS treated samples). (**C, D**) EGF (10 ng/ml) enhanced DNA repair in MCF-7 (**C**) and MB-231 cells (**D**) treated with ETO (10 μM). The Quantification of relative WB band are represented as the mean ± SD of three biological repeats. (**E**) The GFP+ frequency of HR reporter assay in the treatment with EGF (10 ng/ml for 4 hr) or vehicle. Data are expressed as the mean ± standard deviation (SD), n=3 per group. (**F, G**) EGF augmented RAD51 mRNA (**F**) and protein (**G**) expression in MCF-7 cells. Data are expressed as the mean ± standard deviation (SD), n=2 per group. ** p<0.01 (Student's *t*-test).

The online version of this article includes the following source data and figure supplement(s) for figure 4:

**Source data 1.** Source data for *Figure 4A*.

**Source data 2.** Source data for *Figure 4B*.

**Source data 3.** Source data for *Figure 4C*.

**Source data 4.** Source data for *Figure 4D*.

**Source data 5.** Source data for *Figure 4G*.

**Figure supplement 1.** EGF promotes RAD51 expression and enhances homologous recombination repair (HR) efficiency.

**Figure supplement 1—source data 1.** Source data for *Figure 4—figure supplement 1F*.

**Figure supplement 1—source data 2.** Source data for *Figure 4—figure supplement 1G*.

compared with that in control cells (*Figure 6D*), whereas the interaction between YTHDC1 and EGF transcripts was down-regulated in the condition of shMETTL3 (*Figure 6—figure supplement 1D*). Furthermore, the YTHDC1 deficiency impaired the outcome of down-regulation of γ-H2AX foci in METTL3-OV MCF-7 cells (*Figure 6E*). An MTT assay revealed that YTHDC1 depletion rendered MCF-7

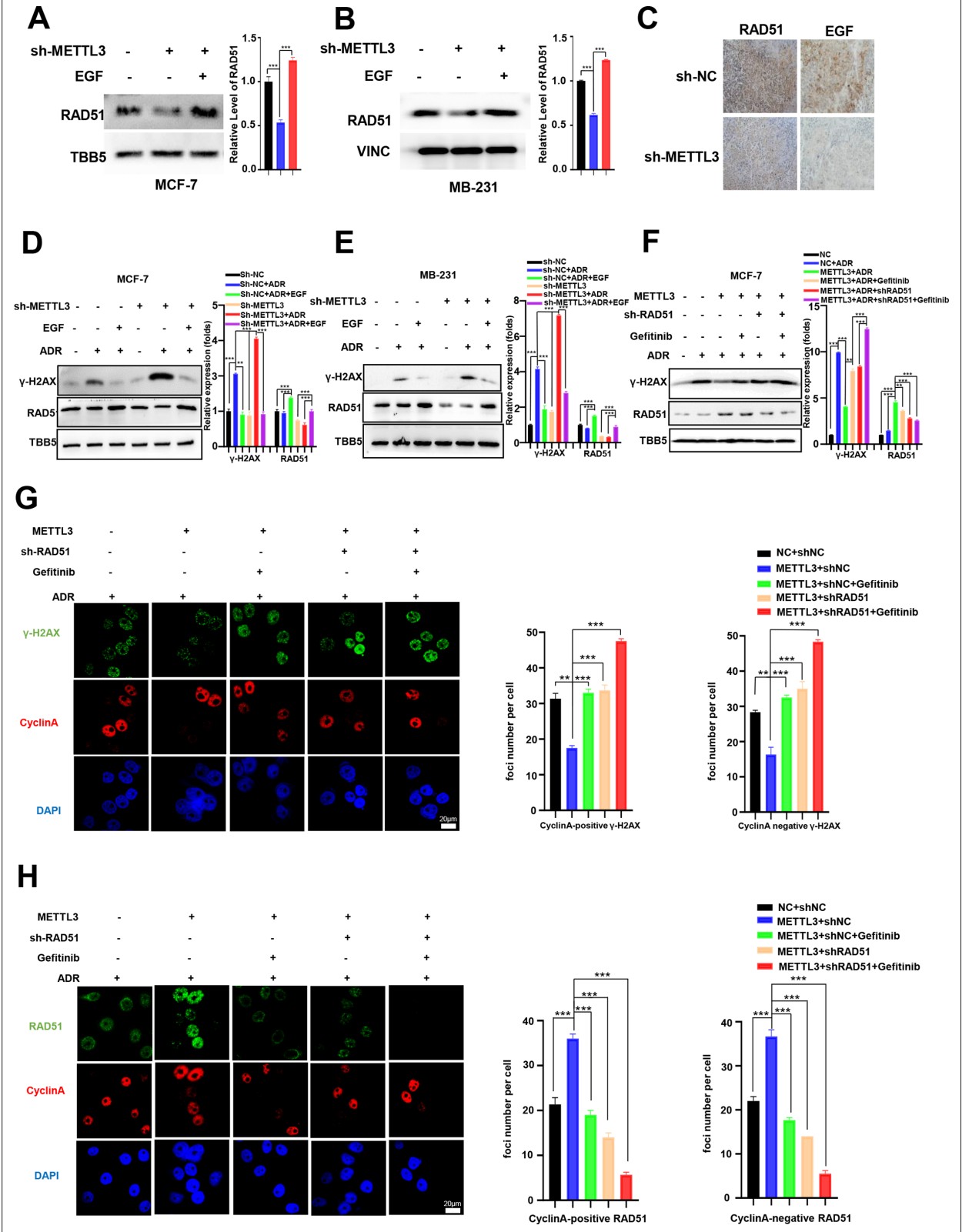

**Figure 5.** Methyltransferase-like 3 (METTL3) promotes DNA repair via EGF/Rad51 axis. (**A, B**) RAD51 protein levels in METTL3-KD MCF-7 (**A**) and MB-231 (**B**) cells treated with or without EGF. (**C**) Immunohistochemistry analysis of the expression of EGF and RAD51 in control and METTL3-KD tumor tissues. (**D, E**) WB analysis showing that treatment with 10 ng/ml EGF for 8 hr restores DNA repair activity in METTL3-KD MCF-7 (**D**) and MB-231 (**E**) cells. (**F**) WB analysis showing that knocking down RAD51 or EGFR inhibitor Gefitinib (10 nM for 8 hr) treatment in METTL3-OV cells decreases

*Figure 5 continued on next page*

*Figure 5 continued*

METTL3-enhanced DNA repair activity. The Quantification of relative WB band are represented as the mean ± SD of three biological repeats. (**G**) Immunofluorescence analysis of co-staining of γ-H2AX and cyclin A in METTL3-OV cells ±RAD51 shRNA or gefitinib (10 nM) during ADR treatment (0.5 μM for 1 hr and recovery for 8 hr without ADR). The quantitative assay is on the right. n=3 per group. (**H**) Immunofluorescence analysis of cyclin A and RAD51 foci in cells treated the same as in (**G**). The quantification of average foci numbers per cells were showed in right panel, 50 cells were calculated in each group. ** p<0.01; *** p<0.001 (Student's *t*-test).

The online version of this article includes the following source data and figure supplement(s) for figure 5:

**Source data 1.** Source data for *Figure 5A*.

**Source data 2.** Source data for *Figure 5B*.

**Source data 3.** Source data for *Figure 5D*.

**Source data 4.** Source data for *Figure 5E*.

**Source data 5.** Source data for *Figure 5F*.

**Figure supplement 1.** METTL3-mediated DNA repair is EGF/ Rad51 dependent.

**Figure supplement 1—source data 1.** Source data for *Figure 5—figure supplement 1A*.

**Figure supplement 1—source data 2.** Source data for *Figure 5—figure supplement 1B*.

**Figure supplement 1—source data 3.** Source data for *Figure 5—figure supplement 1C*.

**Figure supplement 1—source data 4.** Source data for *Figure 2—figure supplement 1D*.

**Figure supplement 1—source data 5.** Source data for *Figure 5—figure supplement 1G*.

**Figure supplement 1—source data 6.** Source data for *Figure 5—figure supplement 1K*.

**Figure supplement 1—source data 7.** Source data for *Figure 2—figure supplement 1L*.

cells more sensitive to ADR and reversed METTL3-induced ADR resistance in METTL3-overexpressing cells (*Figure 6F*). These results were verified by morphological analysis (*Figure 6G*). Moreover, cell survival assay identified that double knockdown of METTL3 and YTHDC1 enhanced the sensitivity of MCF-7 to ADR, which is similar to those of single knockdown of METTL3 or YTHDC1 (*Figure 6— figure supplement 1E*). This result indicated an epistatic effect between shMETTL3 and shYTHDC1 on ADR response in MCF-7 cells. Taken together, our results suggest that YTHDC1 functions as an m6A 'reader' to enhance EGF mRNA stability and augment HR and cell survival in ADR-treated BC cells.

## Discussion

A deficiency of DNA repair proteins is associated with carcinogenesis and elevated DNA repair activity contributes to drug resistance in cancer (*Li et al., 2021*; *Lu et al., 2020*). MELLT3 and its regulated m6A modification have been reported to be involved in DNA repair (*Xiang et al., 2017*; *Zhang et al., 2020*). However, the potential mechanism of METTL3 in DNA repair and chemotherapeutic response is poorly defined. In the present study, we demonstrated that m6A RNA methylation levels and METTL3 expression were elevated in five different kinds of BC cells. Biochemical and cell biological analysis revealed that inhibition of METTL3 sensitizes both ER-positive and triple-negative BC cells to ADR treatment with elevated DNA damage. Knockdown of METTL3 impaired HR activity and increased ADR-induced DNA damage through modification of the EGF/RAD51 axis. METTL3 promoted HR through m6A-dependent upregulation of EGF expression, which further augmented RAD51 expression. The m6A 'reader,' YTHDC1, bound to the m6A-modified EGF transcript, protected EGF mRNA, and enhanced EGF expression (*Figure 7*). This result was consistent with other studies showing that YTHDC1 is recruited to sites of DNA damage, bound m6A RNA, and increased the activity of DSB repair (*Yu et al., 2021a*; *Zhang et al., 2020*).

Recently, RNA m6A modification and the core RNA methyltransferase, METTL3, were reported to play an important role in cancer chemotherapy (*Deng et al., 2018*). METTL3 was implicated in ADR resistance in MCF-7 cells by regulating the miR-221–3 p/HIPK2/Che-1 axis (*Pan et al., 2021*). Our data further demonstrated that METTL3 knockdown markedly sensitized both MCF-7 and MB-231 cells to ADR. ADR is one of the most effective antitumor agents for BC treatment, although it is limited by severe side effects. Various mechanisms have been proposed to explain ADR-induced cell death, including trapping topoisomerase II, formation of ADR-DNA adducts, and generation of free radicals

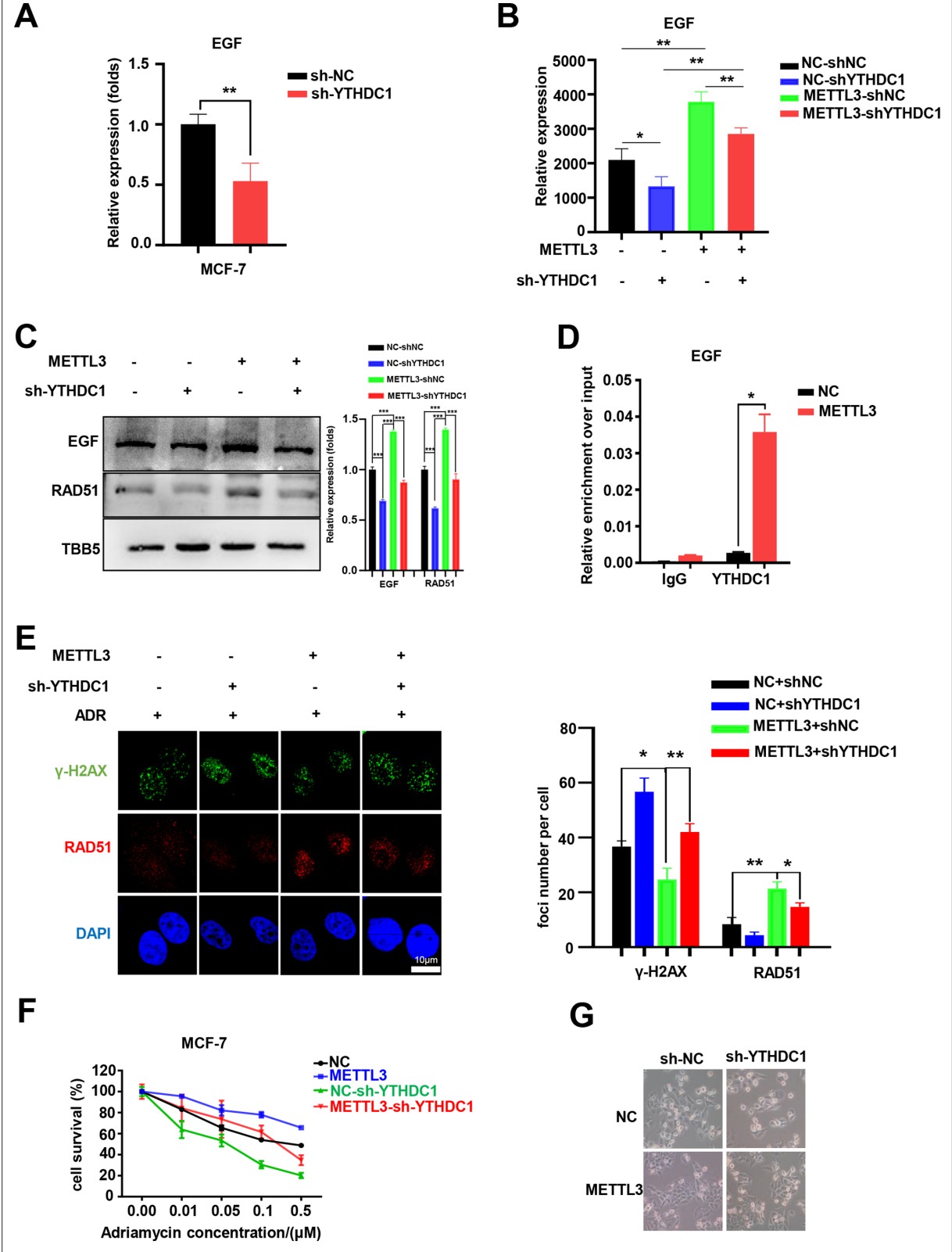

**Figure 6.** YTHDC1 is the 'reader' of the METTL3/m$^6$A-regulated EGF/RAD51 axis. (**A**) The expression of EGF in YTHDC1-silenced MCF-7 cells was detected by qRT-PCR. Data are expressed as the mean ± standard deviation (SD), n=3 per group. (**B**) The mRNA levels of EGF in control or METTL3-OV cells with or without knocking down YTHDC1. Data are expressed as the mean ± standard deviation (SD), n=3 per group. (**C**) WB assay determining the effect of YTHDC1 knockdown on EGF and RAD51 expression in control and METTL3-OV MCF-7 cells. The Quantification of relative WB band are

*Figure 6 continued on next page*

*Figure 6 continued*

represented as the mean ± SD of three biological repeats. (**D**) RIP-qPCR assay showing the enrichment of the EGF transcript in METTL3-OV cells. Data are expressed as the mean ± SD, n=3 per group. (**E**) Immunofluorescence analysis of γ-H2AX and RAD51 foci in METTL3-OV cells with knocked-down YTHDC1. The quantification of average foci numbers per cell are shown in the right panel, 50 cells were calculated in each group. (**F**) MTT assays were performed to detect the effect of YTHDC1 knockdown on ADR sensitivity in control and METTL3-OV MCF-7 cells. Data are expressed as the mean ± standard deviation (SD), n=3 per group. (**G**) Morphological analysis of control or METTL3-OV MCF-7 cells with or without YTHDC1 knockdown. Cells were treated with 0.5 µM ADR for 24 hr. * p<0.05; ** p<0.01; *** p<0.001 (Student's *t*-test).

The online version of this article includes the following source data and figure supplement(s) for figure 6:

**Source data 1.** Source data for *Figure 6C*.

**Figure supplement 1.** YTHDC1 was the reader of METTL3/m6A-regulated EGF.

**Figure supplement 1—source data 1.** Source data for *Figure 6—figure supplement 1F*.

**Figure supplement 1—source data 2.** Source data for *Figure 6—figure supplement 1G*.

**Figure supplement 1—source data 3.** Source data for *Figure 6—figure supplement 1H*.

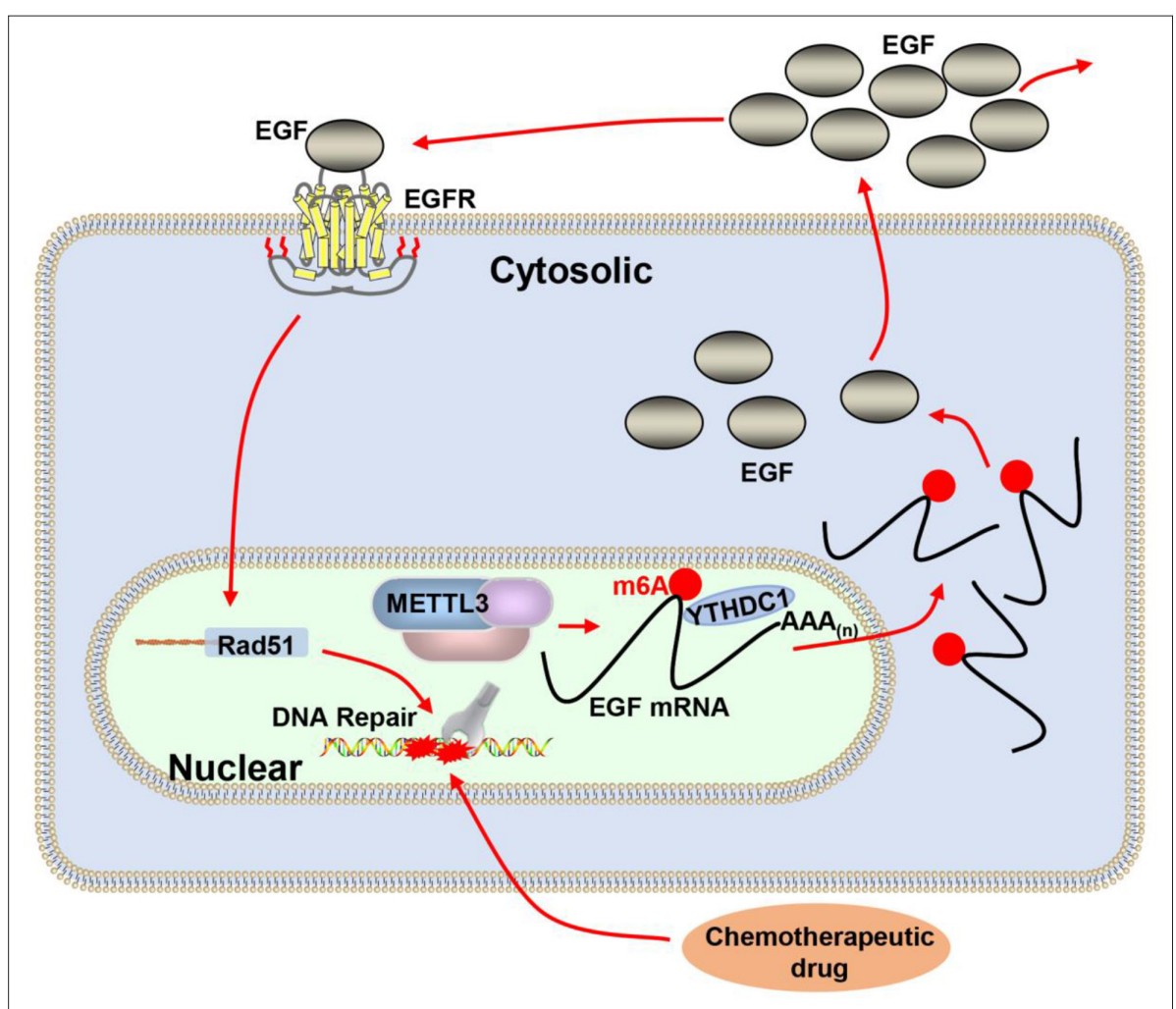

**Figure 7.** Proposed schematic diagram of the proposed mechanism elucidated in this study. Methyltransferase-like 3 (METTL3) augments EGF transcript m6A modification, which was recognized by the reader YTHDC1, resulting increase of EGF expression. Elevated EGF promoted RAD51 expression, enhanced homologous recombination repair (HR) efficacy, and modified chemotherapeutic response of cancer cells.

that increase oxidative stress, which induce DNA damage and result in cell death (*Yang et al., 2014*). We also showed that a deficiency in METTL3 inhibits DNA repair and increases the accumulation of DNA damage in ADR-treated MCF-7 and MB-231 cells, which further leads to cell death.

A large number of studies have implicated dysfunctional DNA repair proteins, such as BRCA1, RAD51, FEN1, and Polβ, in BC initiation and progression (*Li et al., 2021*; *Lu et al., 2020*; *Martin et al., 2007*; *Thacker, 2005*; *Wang et al., 2019*; *Wang et al., 2020b*; *Xia et al., 2021*). Elevated DNA repair activity contributes to drug resistance and limits the efficacy of chemotherapeutic agents (*Long et al., 2021*; *Lu et al., 2020*). Thus, targeting the DDR key proteins such as BRCA1 and BRCA2 is a potential effective therapeutic strategy for mono- or combination therapy. In 2005, the synthetic lethality between PARP inhibition and BRCA1 or BRCA2 mutation was reported by two independent groups, suggesting a novel strategy for treating patients with BRCA-mutant tumors (*Bryant et al., 2005*; *Farmer et al., 2005*; *Xia et al., 2021*). Inhibition of PAPR1 causes failure of DNA single-stranded breaks repair, finally leading to DSBs in cells. HR-competent cells can repair DSBs for cell survival, whereas in cells with defects caused by mutations in BRCA1 or BRCA2 or other HR-associated proteins, the DNA damage remains unrepaired, causing cell death (*Xia et al., 2021*). In 2014, PARP inhibitor Olaparib was primary approved by the FDA to treat certain patients with ovarian cancer. The PARP inhibitors are now indicated for the treatment of patients with germline breast cancer susceptibility gene (BRCA) mutated, human epidermal growth factor receptor 2 (HER2)-negative metastatic BC, who have been previously treated with chemotherapy (*Arora et al., 2021*; *Xia et al., 2021*). Although PARPi has shown great efficacy, their widespread use is restricted by various factors, including drug resistance and the limited population. In recent years, there have been a number of studies and clinical trials evaluating the use of cytotoxic chemotherapy such as temozolomide, platinum salts, and topoisomerase inhibitors in combination with PARPi, and the evidence suggests that combination therapy may be of considerable use in many types of cancer (*Palleschi et al., 2021*). RNA m$^6$A modification has been reported to be involved in the regulation of DNA repair. *Xiang et al., 2017* reported that METTL3-mediated m$^6$A RNA accumulates at UV-damaged sites in DNA, which further recruits DNA polymerase κ (Pol κ) to damaged sites to facilitate NER and cell survival (*Xiang et al., 2017*). Zhang et al., reported METTL3-m$^6$A-YTHDC1 mediated HR in U2OS cells exposed to X-rays or treatment with Zeocin (*Zhang et al., 2020*). Another recent study found that METTL3-METTL14 is active in vitro on double-stranded DNA containing a cyclopyrimidine dimer, and the m$^6$A reader, YTHDC1, is recruited to sites of DNA damage (*Yu et al., 2021a*). Our data demonstrated that silencing METTL3 inhibited the repair of DSBs induced by ADR. Nevertheless, the molecular mechanism of DNA repair regulation by METTL3/m$^6$A remains undefined.

There are two major pathways to repair DSBs including error-free HR and error-prone NHEJ (*Sonoda et al., 2006*). Using a GFP-based reporter system, we found that METTL3 regulated HR, but not NHEJ in DSB repair, which is consistent with the study of *Zhang et al., 2020*. However, Gene Ontology (GO) analysis of METTL3-associated gene expression profiles based on RNA-seq in MCF-7 cells did not identify enriched 'DNA damage' or 'DNA repair' categories (data not shown). This data is supported by Xiang et al., showing that GO analysis of METTL3-dependent, UV-induced methylated transcripts only identified 'DNA damage' as weakly enriched (*Xiang et al., 2017*). Zhang et al., also reported that METTL3 does not change active RNA polymerase II binding to DNA lesions and has no effect on RNA transcription at DSBs. Another study showed that METTL3 catalyzes m$^6$A modification of FEN1 mRNA, which is recognized and stabilized by IGF2BP2 in hepatocellular carcinoma cells (*Pu et al., 2020*). However, our RNA-seq data showed no change in FEN1 expression with METTL3 overexpression in MCF-7 cells, suggesting that METTL3-mediated FEN1 expression may be cell specific.

EGF/EGFR signaling is important to many biological processes, such as cell proliferation, cell division, and tissue development. Human cancer tissues express high levels of growth factors and their receptors, such as EGF/EGFR, which exhibit autocrine or paracrine regulation of cancer growth (*Knowlden et al., 2003*; *Mendelsohn and Baselga, 2000*; *Singh and Harris, 2005*). Aberrant activation of EGF/EGFR signaling contributes to cancer proliferation, epithelial-mesenchymal transition, and metastasis (*Mendelsohn and Baselga, 2000*). Therefore, targeting EGFR by antibodies or small molecule tyrosine kinase inhibitors has been used successfully to treat various malignancies, including BC (*Barzegar et al., 2017*; *Ueno and Zhang, 2011*). Based on RNA-seq, MeRIP-qPCR, and the screening strategy, we determined that EGF is regulated in a METTL3-m$^6$A-YTHDC1-dependent manner in MCF-7 and MB-231 cells. Overexpression of METTL3 markedly increased EGF expression

and secretion, whereas knockdown of METTL3 resulted in the opposite effect. Moreover, we detected the expression of EGFR in METTL3-OV MCF-7, MB-231, and MCF-10A cells. We found that overexpression of METTL3 enhanced the expression of EGFR in all these cells (*Figure 6—figure supplement 1F-H*), which is consistent to previous study (*Lin et al., 2016*). These data indicated METTL3 might have more global effect on EGR-RAD51 axis. We further demonstrated that EGF/EGFR axis plays an important role in METTL3-mediated HR in both ER-positive and triple-negative cells during ADR treatment. Our results are consistent with previous reports showing that EGF/EGFR signaling plays a role in DSB repair (*Myllynen et al., 2011*). Moreover, dysregulated EGF/EGFR signaling modulates the expression of several genes involved in DNA repair including ERCC1, XRCC1, RAD51, and RAD50 (*Kryeziu et al., 2013*; *Yacoub et al., 2003*). Accordingly, we found that EGF/EGFR signaling regulated RAD51 expression in both MCF-7 and MB-231 cells and the regulation of RAD51 expression by METTL3 was EGF-dependent. Furthermore, knockdown of RAD51 or inhibition of EGFR by gefitinib/erlotinib impaired the effect of METTL3 on the up-regulation of DNA repair activity. Although other studies and our RNA-seq data suggest that there may be other factors involved in the regulation of METTL3-mediated DNA repair and response to chemotherapy (*Cai et al., 2018*; *Pan et al., 2021*; *Wang et al., 2020a*). Our experiments demonstrate a role for EGF in the regulation of HR activity and ADR sensitivity in BC.

The m6A modification of EGF mRNA has been detected in human kidney and embryonic stem cells by different groups using m6A-REF-seq (GSE125240) and MAZTER-seq (GSE122961), respectively (*Garcia-Campos et al., 2019*; *Zhang et al., 2019*). Our MeRIP-qPCR assay identified that m6A modification of EGF mRNA was augmented in METTL3-expressing MCF-10A, MCF-7, and MB-231 cells compared with that in control cells. The m6A-modified EGF mRNA was recognized and bound by YTHDC1, which further promoted EGF synthesis and secretion. We demonstrated that the regulation of EGF expression by METTL3 was m6A-YTHDC1-dependent and knockdown of YTHDC1 impeded the up-regulation of EGF in METTL3-overexpressing cells, and restored the sensitivity of MCF-7 cells to ADR. Moreover, knockdown of YTHDC1 impeded METTL3-enhanced RAD51 expression and inhibited recruitment of RAD51 to damaged sites. Our data combined with the studies of Zhang et al., and Yu et al., demonstrate that YTHDC1 plays an important role in DNA repair (*Yu et al., 2021a*; *Zhang et al., 2020*). Our results further suggest the involvement of YTHDC1 in the regulation of EGF mRNA stability. Although YTHDC1 is primary known as a nuclear m6A reader, which regulates mRNA splicing through the recruitment and modulation of pre-mRNA splicing factors such as SRSF3 (*Xiao et al., 2016*). Our data suggest that YTHDC1 may contribute to the stability of cytosolic mRNA, including EGF, through m6A. Our hypothesis is supported by previous studies suggesting that YTHDC1 can shuttle between the nucleus and the cytoplasm (*Rafalska et al., 2004*), to process mature mRNAs, including MAT2A (*Shima et al., 2017*). Moreover, Roundtree et al., showed that YTHDC1 mediates the export of methylated mRNA from the nucleus to the cytoplasm, resulting in nuclear clearance of mRNAs and accompanying their resulting cytoplasmic abundance (*Roundtree et al., 2017*). These results suggest multiple processes through which YTHDC1 regulates the processing of mature mRNA, whereas the detailed molecular mechanism warrants further study.

Collectively, we showed an effect of METTL3 on HR via the m6A-YTHDC1-dependent regulation of the EGF/RAD51 axis, and demonstrated a role for METTL3 in the response of BC chemotherapy. Our results suggest that the development of METTL3 inhibitors or targeting its pathway may lead to promising treatments for cancer patients.

# Materials and methods

## Key resources table

| Reagent type (species) or resource | Designation | Source or reference | Identifiers | Additional information |
|---|---|---|---|---|
| Genetic reagent (*Mus. musculus*) | BALB/c-Nude (BALB/cNj-Foxn1nu/Gpt) | GemPharmatech Co., Ltd., Nanjing, China | Strain NO.D000521 | |
| Cell line (*Homo sapiens*) | MCF-7 | National Collection of Authenticated Cell Cultures, Chinese Academy of Science | CSTR:19375.09. 3101HUMSCSP531 | |

*Continued on next page*

*Continued*

| Reagent type (species) or resource | Designation | Source or reference | Identifiers | Additional information |
|---|---|---|---|---|
| Cell line (*Homo sapiens*) | MDA-MB-231 | National Collection of Authenticated Cell Cultures, Chinese Academy of Science | CSTR:19375.09. 3101HUMSCSP5043 | |
| Antibody | anti-METTL3 (Mouse monoclonal) | ABclonal | Cat# A19079 RRID: Addgene_101892 | WB (1:1000) |
| Antibody | anti-γ-H2AX (Mouse monoclonal) | Cell Signaling Technology | Cat# 80,312 S | WB (1:1000) IF(1:300) |
| Antibody | anti-BAX (Rabbit polyclonal) | ABclonal | Cat# A11550 RRID: AB_516294 | WB (1:1000) |
| Antibody | anti-caspase3 (Rabbit polyclonal) | Proteintech | Cat# 19677-I-AP RRID: AB_590739 | WB (1:1000) |
| Antibody | anti-EGF (Rabbit polyclonal) | Proteintech | Cat# 27141–1-AP RRID: AB_1066833 | WB (1:1000) |
| Antibody | anti-RAD51 (Rabbit polyclonal) | Proteintech | Cat# 14961–1-AP RRID: AB_10706869 | WB (1:1000) IF (1:300) |
| Antibody | anti-FLAG (Mouse monoclonal) | bioworld | Cat# AP0007MH **RRID : AB_1537400** | WB (1:1000) |
| Antibody | anti-EGFR (Rabbit polyclonal) | ABclonal | Cat# A11577 RRID: AB_442085 | WB (1:1000) |
| Antibody | anti-TBB5 (Mouse monoclonal) | Abgent | Cat# AM1031A RRID: AB_1554765 | WB (1:1000) |
| Antibody | anti-Cyclin A (Mouse monoclonal) | proteintech | Cat# 66391–1-Ig | IF(1:300) |
| Sequence-based reagent | METTL3_F | This paper | qPCR primers | AAGCTGCACT TCAGACGAAT |
| Sequence-based reagent | METTL3_R | This paper | qPCR primers | GGAATCACC TCCGACACTC |
| Sequence-based reagent | EGF_F | This paper | qPCR primers | TGGATGTGCTT GATAAGCGG |
| Sequence-based reagent | EGF_R | This paper | qPCR primers | ACCATGTCCTT TCCAGTGTGT |
| Sequence-based reagent | RAD51_F | This paper | qPCR primers | CAACCCATTTC ACGGTTAGAGC |
| Sequence-based reagent | RAD51_R | This paper | qPCR primers | TTCTTTGGCGC ATAGGCAACA |
| Commercial assay or kit | Human EGF ELISA Kit | SenBeiJia Biological Technology Co. | Cat# SBJ-H0212 | |
| Chemical compound, drug | Doxorubicin (Adriamycin) HCl | Selleck | Cat# S1208 CAS No. 25316-40-9 | |
| Chemical compound, drug | Paclitaxel | Selleck | Cat# S1150 CAS No. 33069-62-4 | |
| Chemical compound, drug | Cisplatin | Selleck | Cat# S1166 CAS No. 15663-27-1 | |
| Chemical compound, drug | 5-Fluorouracil, 5-FU | Selleck | Cat# S1209 CAS No. 51-21-8 | |
| Chemical compound, drug | Recombinant Human EGF | Beyotime | Cat# P5552 | |
| Chemical compound, drug | Erlotinib | Beyotime | Cat# SC0168 | |
| Chemical compound, drug | Gefitinib | Beyotime | Cat# SC0186 | |

| Reagent type (species) or resource | Designation | Source or reference | Identifiers | Additional information |
|---|---|---|---|---|
| Software, algorithm | GraphPad Prism software | GraphPad Prism (https://graphpad.com) | RRID: SCR_015807 | Version 8.0.0 |

## Plasmid construction

The oligonucleotide 5'-CAGGAGATCCTAGAGCTATTA-3' was used for construction of METTL3-KD lentivirus vectors as previously described (*Xiang et al., 2017*). The lentivirus vectors were constructed and purified by the Corues Biotechnology Company (Nanjing, China). For knockdown of RAD51, YTHDC1 and other 'readers,' and the silencing plasmids containing shRNA sequences were constructed based on psilencer3.0-H1. The shRNA sequences are listed in *Table 3*. All plasmids were verified by sequencing.

## Cell culture and the development of stable cell lines

MCF-7 and MB-231 were purchased from the National Collection of Authenticated Cell Cultures, Chinese Academy of Science. All cells were authenticated by STR profiling and tested for mycoplasma contamination. Cells were cultured in the recommended medium supplemented with 10% fetal bovine serum (FBS, Invigentech), 1% penicillin, and 1% streptomycin, and incubated in an incubator with 5% $CO_2$ at 37°C. For METTL3-overexpressing (OV) or –knockdown (KD) MCF-7 and MB-231 stable cells, the cells were infected with specific lentivirus vectors for 48 hr and then selected with puromycin for two weeks. All cell lines were confirmed to be negative for mycoplasma contamination.

## $^{m6}$A dot blotting

Total RNA was isolated using the Trizol method and mRNAs were isolated with the GenElute mRNA Miniprep Kit (Sigma). The concentration and purity of the mRNA were measured using a NanoDrop 2000. The mRNAs were denatured by heating to 95°C for 5 min, followed by chilling on ice. Next, the mRNAs (50~100 ng) were spotted directly onto a positively-charged nylon membrane (GE Healthcare, USA) and air-dried at room temperature for 5 min. The membrane was then ultraviolet (UV) crosslinked using a Ultraviolet Crosslinker, washed with PBST for 5 min, blocked with 5% nonfat milk in TBST, and then incubated with anti-$m^6$A antibody (A17924, ABclonal) overnight at 4°C. HRP-conjugated anti-rabbit IgG secondary antibody was added to the membrane for 1.5 hr at room temperature with gentle shaking, followed by development with enhanced chemiluminescence. Last, 0.02% methylene blue staining was used to verify that equal amounts of mRNA were spotted onto the membrane.

## Drug sensitivity assay

Cells were seeded into 96-well plates at 3000 cells per well for at least three parallel experiments. 24 hr later, cells were exposed to ADR at increasing concentrations (0, 0.01, 0.05, 0.1, 0.5, and 1 μM)

**Table 3.** Sequences of the shRNA used in this study.

| Gene name | shRNA sequences |
|---|---|
| sh-METTL3 | 5'-CAGGAGATCCTAGAGCTATTA-3' |
| sh-RAD51 | 5'-GACTGCCAGGATAAAGCTT-3' |
| sh-YTHDC1 | 5'-CCAGAGAGTGAACAAGATAAA-3' |
| sh-YTHDF1 | 5'-GGGGGGTTGAGTGTTGCATCTT-3' |
| sh-YTHDF2 | 5'-AAGGCTAAGCAGGTGTTGAAA-3' |
| sh-YTHDF3 | 5'-TAAGTCAAAGAAGACGTATTACTC-3' |
| sh-YTHDC2 | 5'-GCCCACAGATTGGCTTATTTA-3' |
| sh-IGF2BP1 | 5'-TGCTATTCTTCCTAATCTATATC-3' |
| sh-IGF2BP2 | 5'-GTGAAGCTGGAAGCGCATATCTC-3' |
| sh-IGF2BP3 | 5'-CGGTGAATGAACTTCAGAATTCTC-3' |

for 48 hr; 5-FU at increasing concentrations (0, 0.1, 0.5, 1, 5 µM) for 48 hr; carboplatin at increasing concentrations (0, 10, 50, 100, 500 µM) for 48 hr; paclitaxel at increasing concentrations (0, 0.01, 0.1, 1 nM) for 48 hr; or DDP at increasing concentrations (0, 2, 4, 6, 8 µM) for 48 hr. Chemotherapeutic drug-treated cells were incubated with 10 µL 3-(4,5)-dimethylthiazol (-z-y1)–3,5-diphenyltetrazolium bromide (MTT) solution (5 mg/mL, Sigma-Aldrich, St Louis, MO, USA) for 4 hr. The media was replaced with 100 µL dimethyl sulfoxide (DMSO, Sigma-Aldrich) to dissolve the formazan crystals within 10 min. The absorbance of the formazan was read at 450 nm. The relative values were calculated based on the mean of control cells in the absence of ADR, which was showed as 100%. At least three replicates were performed for each drug treatment.

## RNA-seq and analysis

RNA-Seq was performed by oeBiotech Inc (Shanghai, China). For RNA sequencing, purified RNA from METTL3 overexpressing or control cells was used for library construction with the Illumina TruSeq RNA Sample Prep Kit (FC-122–1001) and then sequenced with an Illumina HiSeq 2000. Raw reads were aligned to the human genome, GRCh37/hg19, by Bowtie2. Differentially expressed genes (DEGs) between METTL3-OV and the control samples were identified using the limma-voom method. A heatmap clustered by k-means was used to show DEGs or transcripts. The raw sequencing data were deposited in the Gene Expression Omnibus database (accession to cite for these SRA data: PRJNA743152).

## RNA immunoprecipitation (RIP)

RNA immunoprecipitation (RIP) assays were conducted using the EZ-Magna RIP RNA-Binding Protein Immunoprecipitation Kit Merck Chemicals (Shanghai Co., Ltd). The anti-YTHDC1 antibody for RIP was purchased from Cell Signaling Technology, Inc (# 81,504 S).

## m$^6$A-RNA immunoprecipitation (MeRIP) and MeRIP-qPCR

m$^6$A enrichment followed by qRT-PCR was used to quantify the changes in m$^6$A methylation of the target gene using the Magna MeRIP m$^6$A Kit (Millipore, MA) following the manufacturer's instructions. Briefly, 5 µg of fragmented mRNA extracted from MCF-7 stable cells was incubated with 5 µg of m$^6$A antibody (A17924, ABclonal). Methylated mRNA was eluted by free m$^6$A from the beads and purified with the GenElute mRNA Miniprep Kit (MRN70, Sigma). One tenth of the fragmented RNA was saved as an input control for standardization. The relevant enrichment of m$^6$A from METTL3 in each sample was analyzed by RT-qPCR.

## Immunofluorescence

For immunofluorescence assays, cells were washed with PBS for three times then fixed with 4% formaldehyde for 10 min at room temperature. After permeabilization with 0.1% Triton X-100 for 10 min, cells were blocked with 3% BSA for 1 hr. Then, cells were incubated with indicated primary antibodies overnight at 4 °C. Following washed with PBST for three times, cells were incubated with fluorescent secondary antibodies for 2 hr at room temperature. Subsequently, cells were stained with DAPI and visualized under a fluorescence microscope (Nikon 80I 10–1500×).

## Tumor analysis

All animal experiments were performed according to the procedures approved by the Laboratory Animal Care Committee at Nanjing Normal University (Permit number IACUC—20210251) and followed National Institutes of Health guide for the care and use of Laboratory animals. Six-to-seven weeks old female nude mice were purchased from GemPharmatech Co., Ltd. (Nanjing, China), and were maintained under specific pathogen-free conditions for subcutaneous inoculation. Cells were trypsinized and resuspended in DMEM at a consistence of $1 \times 10^7$ cells/ml. A total of $1 \times 10^6$ cells were injected into flank of mice. 27 days after injection, tumors were removed for paraffin-embedded sections.

## Immunohistochemistry

Immunohistochemical staining was performed as previously described (*Lu et al., 2020*). Briefly, tumor tissues were fixed in 4% polysorbate. Paraffin-embedded sections from tissue specimens were

deparaffinized and heated at 100 °C in 10 mM citrate buffer (pH 6.0) for 15 min for antigen retrieval. Slides were incubated with primary antibody at 4 °C overnight, followed by incubation with secondary antibody at room temperature and visualized using a DAB Kit (Bioworld). Then, it was redacted with hematoxylin. The expression levels of target proteins in tissue were examed according to the semi-quantitative immunoreactivity score (IRS).

## Apoptosis assay

METTL3-KD or control MCF-7 or MB-231 cells were treated with doxorubicin for 24 hr and replaced with fresh media. After other 24 hr recovery, about $1 \times 10^5$ cells per well were collected and stained with both Annexin V and propidium iodide (PI). Apoptosis was analyzed by flow cytometry using the BD FACSverse.

## ELISA

The cell culture media were centrifuged at the speed of 5000 r.p.m for 5 min and supernatant were collected for EGF measurement using commercial kits (SenBeiJia Biological Technology Co., Nanjing, China) according to manufacturer's instructions.

## Statistical analysis

Statistical analysis was performed with GraphPad Prism 8.0. Statistical significance was determined using a two-tailed Student's t-test or analysis of variance in the case of comparisons among multiple groups. $p < 0.05$ was considered statistically significant.

## Additional information

### Funding

| Funder | Grant reference number | Author |
| --- | --- | --- |
| National Natural Science Foundation of China | 32171407 | Zhigang Hu |
| National Natural Science Foundation of China | 81872284 | Zhigang Guo |
| Natural Science Fund of Colleges and Universities in Jiangsu Province | 19KJA180010 | Zhigang Hu |
| Priority Academic Program Development of Jiangsu Higher Education Institutions | | Zhigang Guo |

The funders had no role in study design, data collection and interpretation, or the decision to submit the work for publication.

### Author contributions

Enjie Li, Data curation, Formal analysis, Investigation, Methodology, Resources, Visualization, Writing - original draft; Mingyue Xia, Conceptualization, Data curation, Methodology, Visualization; Yu Du, Data curation, Formal analysis, Investigation, Validation, Visualization; Kaili Long, Data curation, Formal analysis, Methodology, Visualization, Writing - review and editing; Feng Ji, Methodology, Validation, Visualization; Feiyan Pan, Lingfeng He, Methodology, Resources, Writing - review and editing; Zhigang Hu, Funding acquisition, Investigation, Methodology, Project administration, Supervision, Writing - original draft, Writing - review and editing; Zhigang Guo, Data curation, Funding acquisition, Methodology, Project administration, Resources, Supervision, Writing - review and editing

### Author ORCIDs

Enjie Li http://orcid.org/0000-0002-1113-4384
Yu Du http://orcid.org/0000-0002-1847-0743
Feiyan Pan http://orcid.org/0000-0002-3990-618X

Zhigang Hu http://orcid.org/0000-0002-5265-3535

### Ethics

All animal experiments were performed according to the procedures approved by the Laboratory Animal Care Committee at Nanjing Normal University (Permit number IACUC-20210251) and followed National Institutes of Health guide for the care and use of Laboratory animals.

### Decision letter and Author response

Decision letter https://doi.org/10.7554/eLife.75231.sa1
Author response https://doi.org/10.7554/eLife.75231.sa2

## Additional files

### Supplementary files

• Transparent reporting form

### Data availability

The raw sequencing data were deposited in the Gene Expression Omnibus database (accession to cite for these SRA data: PRJNA743152).

The following dataset was generated:

| Author(s) | Year | Dataset title | Dataset URL | Database and Identifier |
|---|---|---|---|---|
| Enjie L, Zhigang H | 2021 | METTL3 Overexpressing MCF-7 | https://www.ncbi.nlm.nih.gov/bioproject/PRJNA743152 | NCBI BioProject, PRJNA743152 |

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
