## [Editor Report]

This paper identified a new mechanism by which N6-methyltransferase METTL3 modulates DNA repair and cellular resistance to DNA damaging chemotherapeutics. The mechanism involves upregulation of epidermal growth factor expression, which in turn regulates expression of RAD51 recombinase resulting in enhanced DNA repair by homologous recombination.

---

## [Decision Letter]

**Decision letter after peer review:**

Thank you for submitting your article "METTL3 promotes homologous recombination repair and modulates chemotherapeutic response by regulating the EGF/Rad51 axis" for consideration by *eLife*. Your article has been reviewed by 3 peer reviewers, and the evaluation has been overseen by a Reviewing Editor and Kevin Struhl as the Senior Editor. The following individual involved in review of your submission has agreed to reveal their identity: Sarah R Hengel (Reviewer #1).

Essential revisions:

Please respond to the reviewers' critiques. Most importantly, address with the inclusion of necessary experiments.

1) How the cell cycle influences the outcomes of the reporter assays.

2) The effects of the RAD51 overexpression.

3) Monitor ADR resistance in response to METTL3 overexpression.

Additionally, please address all the questions regarding data quantification and analysis

*Reviewer #1 (Recommendations for the authors):*

1. Throughout the text the authors refer to responses in breast cancer (BC) cells. As different types of breast cancers (ER+, TNBC, HER2, BRCA1/2-/-) have different signaling pathways and genetic factors that play an important role, it is a standard in the field to explicitly list the types of breast cancers one is referring to throughout the text. Please address throughout the entire text and figure legends.

2. Each set of figures should have a concluding statement about the larger conclusion drawn from all the data presented in that figure. Some of the figures have this statement but not all. Please address.

3. There are not limitations on references cited for *eLife*. For example, several key references for paragraph (line 59-71) are lacking historical references from within the field with regard to chemotherapeutics, doxyrubicin, etc. Please amend.

4. While ADR is a chemical name used, the more well-known name for this compound is doxorubicin. I would suggest that the authors utilize this name throughout the text, but this is just a suggestion.

5. While topoisomerase inhjibitors were once the gold standard-breast cancers are more often readily treated with PARPi (ex Olaparib) in the clinic in the United States. I think it essential for mechanistic understanding of the interplay of historically important chemotherapeutics (doxyrubicin) and current standards are addressed in this manuscript (Olaparib).

6. Throughout the text there are western blots that should be quantified. This will allow the data to be quantitatively described and compared instead of using non-quantitative terms like "increased" etc.

8. Γ H2AX phosphorylation is a global marker of DSBs and stalled forks. The authors failed to note that H2AX phorylation is present and a marker of stalled replications forks.

a. PMID: 11673449, PMID: 20053681, doi:10.1101/gad.2053211, https://doi.org/10.1016/j.cell.2013.10.043 etc.

*Reviewer #2 (Recommendations for the authors):*

Overall, I found the study and individual experiments underdeveloped, with major points as follows.

1. Throughout the manuscript, cell cycle control should be assessed, which can affect RAD51 foci / HR frequencies. For example (Figure 5G, H), RAD51 foci should be performed with co-staining with either Cyclin A or EdU labeling. Alternatively, examining cell cycle profiles with π / BrdU labeling would be fine. Without such analysis, it is unclear whether all the findings are potentially due to increased proliferation vs. HR proficiency per se.

2. Figure 2/4 concerns

Overall concern: depletion of METTL3 is shown for panels A-F (gammaH2AX dynamics), and METTL3 over-expression for the HR reporter assay. The rationale for not showing data for both manipulations of METTL3 in both end point analysis is unclear.

i) Panels A-D (and Figure 4 A-D), A gammaH2AX time-course is shown only once, but should be repeated and quantified.

ii) Panels E-F. Only one time point of gammaH2AX foci is shown, and untreated cells are also not shown. gammaH2AX foci density can also mark apoptotic cells, and so DNA damage should also be tested with 53BP1 foci; while a single IF representative image is shown in supplemental, this should be quantified.

iii) Panel H (and Figure 4E). The raw GFP+ frequencies should be shown. Normalizing the +METTL3 to 100% is a somewhat confusing way to describe these experiments

3. The significance of elevated RAD51 causing higher HR / more RAD51 foci is underdeveloped. It is unclear whether elevating the levels of RAD51 is actually pro-HR, as the literature is mixed on this point. While the authors perform depletion of RAD51, the appropriate mimic for the +METTL3 or +EGF experiments is over-expression of RAD51. As in point #1, RAD51 levels should be assessed more rigorously with cell cycle phase experiments, and examination of RAD51 cytoplasmic levels.

*Reviewer #3 (Recommendations for the authors):*

1) Figure 1: in figure 1 A the authors mention that in the absence of ADR the viability is 100%. However, in the later blot panel E, it looks that there is some apoptosis even without the drug, which means that the shRNA is toxic. How were these data normalized? Accordingly, this information should be described in detail in material and methods or in figure legends.

2) Figure-1C: On the top of the images there are numbers but without any other information – For instance which drug is used. This information should be present to help future readers to understand the figure. Also, μm should read uM. I have the same comment for the other panels through all the paper ex:

– Figure S1-J- add the drug used at the top of the figure (1uM of what).

– Supplementary Figure 1L, M: please indicate in the figure the dose used for this experiment.

– Supp figure 2 -E and F, indicate the scale bar.

3) Figure S1-J- The authors should consider to overexpress METTL3 and monitor increase to ADR in MCF-10A. Do you increase m6A of the EGF mRNA and the binding of the reader? Do you increase the expression of the EGF? This should be tested to validate the model.

4) Figure 2-A-B. Therefore, it is difficult to make a conclusion without quantification. Please provide the relative quantification of triplicate experiments. This should be compared with the results of the IF.

5) Supplementary Figure S2E-F, the authors should provide the quantification of these figures.

6) Material and methods: Please provide the list of the antibodies used for this study especially for the IF experiments.

7) Figure 3-D: the heat map is not ready to understand. On top: please clarify the meaning of the group highlighted in red and the group highlighted in blue. Does each column represent replicates? I guess in red is represents METLL3? The scale indicates number but what is the value, is it Log10 or Ln2 scaled?

8) Figure 3I: a control is missing in the figure. What is the signal obtained in the condition of shMETTL3 or siRNA?

9) Figure 4E and Supplementary Figure S4A, B: In the text the authors did not describe what they did in these experiments. What is Gefitinib and erlotinib? We can find this information but only later.

10) Figure 5: What are the results of the epistatic test on the viability in the single and the double shRNA conditions shMETLL3 and shEGF in terms of growth and RAD51 foci?

11) Figure 6D: A control is missing, this interaction should be abrogated in the absence of MELLT3.

12) Figure 6F: A control is missing – what is the phenotype of shMETLL3 and shYTHDC1 double knockdown are their epistatic? They should.

13) Discussion: It has been showed previously that METTL3 Promotes EGF receptors translation in Human Cancer Cells. The authors should monitor the expression of the EGFR receptors. This is likely to also enhance the response of the EGR-RAD51 axis. Once the data obtained, this should be added to the discussion (PMID: 27117702).

---

## [Author Response]

Essential revisions:Please respond to the reviewers' critiques. Most importantly, address with the inclusion of necessary experiments.1) How the cell cycle influences the outcomes of the reporter assays.

In this study, we identified that METTL3 mediated HR activity and regulated chemotherapeutic response in breast cancer cells. As HR primarily operates in the S and G2 phases of the cell cycle, cell cycle can affect HR frequencies. We also wonder if METTL3-mediated HR is a direct or indirect effect of the change in cell cycle profile. To address this question and per #2 reviewer’s suggestion, we stained Cyclin A to represent S/G2 phase cells and performed the co-staining of γ-H2AX and RAD51 foci with Cyclin A. We showed γ-H2AX were down-regulated by overexpression of METTL3 in both cyclin A positive cells and cyclin A negative cells (new Figure 5G). Furthermore, the RAD51 foci were augmented by overexpression of METTL3 and reversed with treatment of EGFR inhibitor gefitinib in both cyclin A positive cells and cyclin A negative cells (new Figure 5H). In contrast, the RAD51 foci decreased in cells inhibition of METTL3 by shRNA or inhibition of EGF/EGFR by gefitinib in both cyclin A positive cells and cyclin A negative cells (new Figure 5—figure supplement 1I).

2) The effects of the RAD51 overexpression.

To address this question, we overexpressed RAD51 or knocked down this protein in (GFP)-based HR reporter system to detect the effect of RAD51. Our data showed that knockdown of RAD51 repressed HR activity, whereas overexpression of RAD51 promoted HR efficiency (new Figure 5—figure supplement 1E and F), which were consistent with other current studies (Asan, 2019; Ouyang, 2021). Furthermore, we detectedγ-H2AX levels in the condition of overexpressed RAD51 to mimic +METTL3 or +EGF experiment. We found that overexpression of RAD51 could alleviate γ-H2AX levels, which is similar to the effect of overexpression of METTL3 and EGF (new Figure 5—figure supplement 1K, L).

3) Monitor ADR resistance in response to METTL3 overexpression.

To address this question, we overexpressed METTL3 in MCF-10A cells and detected their response to ADR. Using this cell model, we found that overexpression of METTL3 attenuated the sensitivity of MCF-10A cells to ADR (new Figure 1 —figure supplement 1L-N). Moreover, the m6A modification of EGF mRNA was upregulated in METTL3-OV MCF-10A cells, as well as the binding of EGF to YTHDC1 (new Figure 3 —figure supplement 3J).

Additionally, please address all the questions regarding data quantification and analysis

We quantified all WB and IF data as suggested and point-by-point responses to the comments are attached.

Reviewer #1 (Recommendations for the authors):1. Throughout the text the authors refer to responses in breast cancer (BC) cells. As different types of breast cancers (ER+, TNBC, HER2, BRCA1/2-/-) have different signaling pathways and genetic factors that play an important role, it is a standard in the field to explicitly list the types of breast cancers one is referring to throughout the text. Please address throughout the entire text and figure legends.

In this study, we detected the effect of METTL3 on HR and chemotherapeutic response using both MCF-7 and MDA-MB-231 breast cancer cells. MCF-7 is a human breast cancer cell line with estrogen, progesterone and glucocorticoid receptors. The MCF-7 cell are useful in vitro breast studies because they retained several ideal characteristics particular to mammary epithelium (Camarillo et al., 2014). In contrast, the MDA-MB-231 cell line is ER, PR, and E-cadherin negative, which is a good model of triple-negative breast cancer (Welsh, 2013). We labeled these specific cell lines throughout the text and figure legends in our revised manuscript.

2. Each set of figures should have a concluding statement about the larger conclusion drawn from all the data presented in that figure. Some of the figures have this statement but not all. Please address.

We thank the reviewer for this suggestion. We modified the figure legends of our revised manuscript.

3. There are not limitations on references cited for eLife. For example, several key references for paragraph (line 59-71) are lacking historical references from within the field with regard to chemotherapeutics, doxyrubicin, etc. Please amend.

Thanks for this critique. We added some description and references in the revised version.

4. While ADR is a chemical name used, the more well-known name for this compound is doxorubicin. I would suggest that the authors utilize this name throughout the text, but this is just a suggestion.

Indeed, doxorubicin is the well-known name in the clinical treatment. As we most focus on the effect of ADR on HR and chemotherapeutic response of BC cells, we keep the name ADR in our text. We also indicated another name of ADR is doxorubicin in the revised abstract.

5. While topoisomerase inhibitors were once the gold standard-breast cancers are more often readily treated with PARPi (ex Olaparib) in the clinic in the United States. I think it essential for mechanistic understanding of the interplay of historically important chemotherapeutics (doxyrubicin) and current standards are addressed in this manuscript (Olaparib).

Doxorubicin was the first line chemotherapeutic agent for BC chemotherapy. Various mechanisms have been proposed to explain ADR-induced cell death, including trapping topoisomerase II, formation of ADR-DNA adducts, and generation of free radicals that increase oxidative stress, which induce DNA damage and result in cell death (Yang, 2014).

Targeting the DNA damage response (DDR) key proteins such as BRCA1 and BRCA2 is an effective therapeutic strategy for mono- or combination therapy. In 2005, two groups described the synthetic lethality (SL) interaction between PARP inhibition and BRCA1 or BRCA2 mutation, suggesting a novel strategy for treating patients with BRCA-mutant tumors (Farmer, 2005; Bryant, 2005; Xia, 2021). PARP inhibitors, originally used as chemo or radiosensitizing agents, are effective in the treatment of homologous recombination repair (HRR)-defective tumors. The PARP inhibitor, Olaparib was primary approved by the FDA in 2014 to treat certain patients with ovarian cancer and is now indicated for the treatment of patients with germline breast cancer susceptibility gene (BRCA) mutated, human epidermal growth factor receptor 2 (HER2)-negative metastatic breast cancer, who have been previously treated with chemotherapy (Arora, 2021; Xia, 2021). Although PARPi has shown great efficacy, their widespread use is restricted by various factors, including drug resistance and the limited population. In recent years there have been a number of studies and clinical trials evaluating the use of cytotoxic chemotherapy such as temozolomide,platinum salts, and topoisomerase inhibitors in combination with PARPi, and the evidence suggests that combination therapy may be of considerable use in many types of cancer (Palleschi, 2021). In the revised version we added a description in the Discussion section.

6. Throughout the text there are western blots that should be quantified. This will allow the data to be quantitatively described and compared instead of using non-quantitative terms like "increased" etc.

We thank the comments from the reviewer. We quantified the densities of Western blotting bands and added the data in the revised figures.

8. Γ H2AX phosphorylation is a global marker of DSBs and stalled forks. The authors failed to note that H2AX phorylation is present and a marker of stalled replications forks.a. PMID: 11673449, PMID: 20053681, doi:10.1101/gad.2053211, https://doi.org/10.1016/j.cell.2013.10.043 etc.

We agree with the Reviewer. Γ H2AX is the marker of both DNA damage and DNA replication stress. We apologize that we didn’t mention this in our initial manuscript, and we added the description in the revised version. We also detected and added another DNA damage marker 53BP1 foci in the revised Figure 2F to identify the effect of METTL3 in DNA damage response. In addition, we stained Cyclin A to represent S/G2 phase cells and we showed that γ-H2AX foci decreased in both cyclin A positive cells and cyclin A negative cells with overexpression of METTL3 (new Figure 5G).

Reviewer #2 (Recommendations for the authors):Overall, I found the study and individual experiments underdeveloped, with major points as follows.1. Throughout the manuscript, cell cycle control should be assessed, which can affect RAD51 foci / HR frequencies. For example (Figure 5G, H), RAD51 foci should be performed with co-staining with either Cyclin A or EdU labeling. Alternatively, examining cell cycle profiles with π / BrdU labeling would be fine. Without such analysis, it is unclear whether all the findings are potentially due to increased proliferation vs. HR proficiency per se.

We thank the reviewer for this important comment. As HR primarily operates in the S and G2 phases of the cell cycle, cell cycle can affect RAD51 foci / HR frequencies. Per the reviewer’s suggestion, we performed the co-staining of γ-H2AX and RAD51 foci with Cyclin A. The γ-H2AX foci were down-regulated by overexpression of METTL3, which were reversed by treatment with siRAD51 or EGFR inhibitors in both cyclin A positive cells and cyclin A negative cells (Figure 5G). Consistently, we also found that RAD51 foci were augmented by overexpression of METTL3 and reversed with treatment of EGFR inhibitor gefitinib in both cyclin A positive cells and cyclin A negative cells (new Figure 5H). In contrast, we found that the RAD51 foci decreased in cells inhibition of METTL3 by shRNA or inhibition of EGF/EGFR by gefitinib in both cyclin A positive cells and cyclin A negative cells (Figure 5—figure supplement 1I). We added these data in the revised version.

2. Figure 2/4 concernsOverall concern: depletion of METTL3 is shown for panels A-F (gammaH2AX dynamics), and METTL3 over-expression for the HR reporter assay. The rationale for not showing data for both manipulations of METTL3 in both end point analysis is unclear.

Thanks for this comment. We detected the γ-H2AX levels in condition of both depletion of METTL3 and overexpression of METTL3 (as showed in Figure 2A-D and Figure 2—figure supplement 1A-D). Per the reviewer’s suggestion, we detected the HR activity with knockdown of METTL3 for HR reporter assay in revised Figure 2H. We found that overexpression of METTL3 enhanced HR activity, whereas knockdown of METTL3 decreased efficiency of HR (new Figure 2H; and Figure 2—figure supplement 1G).

i) Panels A-D (and Figure 4 A-D), A gammaH2AX time-course is shown only once, but should be repeated and quantified.

The experiments were repeated for three time and the typical images were selected to show in figures. As per the reviewer’s suggestion, the percentage of positive cell with γ-H2AX foci were quantified and presented in the new Figure 2A-D and Figure 4A-D.

ii) Panels E-F. Only one time point of gammaH2AX foci is shown, and untreated cells are also not shown. gammaH2AX foci density can also mark apoptotic cells, and so DNA damage should also be tested with 53BP1 foci; while a single IF representative image is shown in supplemental, this should be quantified.

We thank the reviewer for these suggestions. We agree that the γ-H2AX was not only a DNA damage marker. Per the reviewer’s suggestion, we tested the 53BP1 foci in the same condition of γ-H2AX detection (new Figure 2F). We also quantified all IF figures and added the data in the revised figures.

iii) Panel H (and Figure 4E). The raw GFP+ frequencies should be shown. Normalizing the +METTL3 to 100% is a somewhat confusing way to describe these experiments

As per the reviewer’s suggestion, we re-organized the data of HR reporter and quantified the data of the GFP signal following Dr Jeremy Stark's lab (Tsai 2020; Mendez-Dorantes, 2020).

3. The significance of elevated RAD51 causing higher HR / more RAD51 foci is underdeveloped. It is unclear whether elevating the levels of RAD51 is actually pro-HR, as the literature is mixed on this point. While the authors perform depletion of RAD51, the appropriate mimic for the +METTL3 or +EGF experiments is over-expression of RAD51. As in point #1, RAD51 levels should be assessed more rigorously with cell cycle phase experiments, and examination of RAD51 cytoplasmic levels.

We appreciate these comments and thank the reviewer for his/her good suggestion. Per the reviewer’s suggestion, we overexpressed and knocked down of RAD51 in (GFP)-based HR reporter system. We found that overexpression of RAD51 promoted HR activity, whereas knockdown of RAD51 repressed HR activity (Figure 5—figure supplement 1E, F), which were consistent with other current studies (Asan, 2019; Ouyang, 2021). Furthermore, we detectedγ-H2AX levels in the condition of overexpressed RAD51 following the reviewer’s suggestion to mimic +METTL3 or +EGF experiment. We found that overexpression of RAD51 could alleviate γ-H2AX levels, which are equal to the effect of overexpression of METTL3 and EGF (Figure 5—figure supplement 1K, L). We added all these supplementary data in revised version.

Reviewer #3 (Recommendations for the authors):1) Figure 1: in figure 1 A the authors mention that in the absence of ADR the viability is 100%. However, in the later blot panel E, it looks that there is some apoptosis even without the drug, which means that the shRNA is toxic. How were these data normalized? Accordingly, this information should be described in detail in material and methods or in figure legends.

We thank these comments from the reviewer. We measured the cell survival by MTT assays and calculated the relative value based on the mean of cells in the absence of ADR, which was showed as 100% (Figure 1A, B). For apoptosis analysis in Figure 1E, the cells with different treatments were collected and stained with both Annexin V and propidium iodide (PI). Then, the apoptosis was analyzed normally by flow cytometry using the BD FACSverse. We have added the data normalization in the revised “Drug sensitivity assay” method.

2) Figure-1C: On the top of the images there are numbers but without any other information – For instance which drug is used. This information should be present to help future readers to understand the figure. Also, μm should read uM. I have the same comment for the other panels through all the paper ex:– Figure S1-J- add the drug used at the top of the figure (1uM of what).– Supplementary Figure 1L, M: please indicate in the figure the dose used for this experiment.– Supp figure 2 -E and F, indicate the scale bar.

Thanks for these suggestions. All these figures were updated as suggested. For ‘Supplementary Figure 1L, M’, the ADR dose and treatment were showed in revised figure legends of new Figure 1 —figure supplement 1O and P.

3) Figure S1-J- The authors should consider to overexpress METTL3 and monitor increase to ADR in MCF-10A. Do you increase m6A of the EGF mRNA and the binding of the reader? Do you increase the expression of the EGF? This should be tested to validate the model.

These are important suggestions. Per the reviewer’s suggestions, we overexpressed METTL3 in MCF-10A cells and detected their response to ADR. Using this cell model, we confirmed that overexpression of METTL3 attenuated the sensitivity of MCF-10A cells to ADR (new Figure 1 —figure supplement 1L-N). Consistently, the m6A modification of EGF mRNA was upregulated in METTL3-overexpressing MCF-10A cells, as well as the binding of the reader (new Figure 3 —figure supplement 3J). We also found that overexpression of METTL3 promoted the expression of EGF in MCF-10A cells (new Figure3 —figure supplement 3G, H). Our supplementary data confirmed the effect of METTL3 on EGF expression and ADR response in MCF-10A cell model.

4) Figure 2-A-B. Therefore, it is difficult to make a conclusion without quantification. Please provide the relative quantification of triplicate experiments. This should be compared with the results of the IF.

As per the reviewer’s suggestion, we quantified the Western blotting band in the revised Figure 2 as well as in other figures.

5) Supplementary Figure S2E-F, the authors should provide the quantification of these figures.

We provided the quantification in these revised figures.

6) Material and methods: Please provide the list of the antibodies used for this study especially for the IF experiments.

We provided the antibodies list in the Key resources table.

7) Figure 3-D: the heat map is not ready to understand. On top: please clarify the meaning of the group highlighted in red and the group highlighted in blue. Does each column represent replicates? I guess in red is represents METLL3? The scale indicates number but what is the value, is it Log10 or Ln2 scaled?

We apologize for making this confusion. Yes, the red represents METTL3. We have modified this panel as suggested. The scale indicates the value of log_2_ (fold change) and we add the description in the revised figure legend of Figure 3.

8) Figure 3I: a control is missing in the figure. What is the signal obtained in the condition of shMETTL3 or siRNA?

We thank the reviewer for this critique. Per the reviewer’s suggestion, we detected the Me-RIP-qPCR of EGF in the condition of shMETTL3. While overexpression of METTL3 upregulated the m6A modification of EGF mRNA, knockdown of METTL3 significantly attenuated its m6A levels (new Figure 3 —figure supplement 1J).

9) Figure 4E and Supplementary Figure S4A, B: In the text the authors did not describe what they did in these experiments. What is Gefitinib and erlotinib? We can find this information but only later.

We apologize for making this confusion. We added some description of the experiments about Figure 4E and Supplementary Figure S4a, b. In these experiments, we tried to detect the effect of the EGF on regulating HR activity using a GFP-based HR reporter system as showed in Figure 2G. We added additional EGF (Figure 4E) or EGFR inhibitor Gefitinib and erlotinib (Supplementary Figure S4A, B) in the Reporter system, and showed that EGF promoted HR, whereas EGFR inhibitor repressed HR, showing by increased or reduced GFP-positive cells.

10) Figure 5: What are the results of the epistatic test on the viability in the single and the double shRNA conditions shMETLL3 and shEGF in terms of growth and RAD51 foci?

We appreciate these comments. We detected RAD51 foci and cell survival in cells with single inhibition of METTL3 by shRNA or inhibition of EGF/EGFR by gefitinib or double treatments with shMETTL3 and gefitinib. We found that the RAD51 foci decreased to similar levels in these three conditions, which indicated an epistatic effect of shMETTL3 and EGF/EGFR inhibition. This epistatic effect was verified by cells survival assay (Figure 5—figure supplement 1I, J). We added these data in the revised manuscript.

11) Figure 6D: A control is missing, this interaction should be abrogated in the absence of MELLT3.

As per the reviewer’s suggestion, we detected the EGF mRNA binding of reader in the cells with knockdown of METTL3. Indeed, our RIP-qPCR experiments showed that down-regulation of METTL3 markedly suppressed the binding of EGF mRNA to YTHDC1 (new Figure 6—figure supplement 1D). We added these data in the revised manuscript.

12) Figure 6F: A control is missing – what is the phenotype of shMETLL3 and shYTHDC1 double knockdown are their epistatic? They should.

As per the reviewer’s suggestion, we tested the cell survival in cells with double knockdown of METTL3 and YTHDC1. We found that cells showed similar sensitivity to ADR treatment in the condition of shYTHDC1 and shMETTL3, as well as in the condition of double knockdown of METTL3 and YTHDC1 (new Figure 6—figure supplement 1E).

13) Discussion: It has been showed previously that METTL3 Promotes EGF receptors translation in Human Cancer Cells. The authors should monitor the expression of the EGFR receptors. This is likely to also enhance the response of the EGR-RAD51 axis. Once the data obtained, this should be added to the discussion (PMID: 27117702).

As per the reviewer’s suggestion, we detected the expression of EGFR in METTL3-OV MCF-7, MB-231 and MCF-10A cells. We found that overexpression of METTL3 also enhanced the expression of EGFR (new Figure 6—figure supplement 1F-H), which is consistent to the data reported by Lin et al. (Lin, 2016). These data indicated METTL3 might has more global effect on EGR-RAD51 axis. We added a discussion in the revised version.

References:

Asan A, Skoko JJ, Woodcock CC, Wingert BM, Woodcock SR, Normolle D, Huang Y, Stark JM, Camacho CJ, Freeman BA, Neumann CA. Electrophilic fatty acids impair RAD51 function and potentiate the effects of DNA-damaging agents on growth of triple-negative breast cells. J Biol Chem. 2019 Jan 11;294(2):397-404.

Arora S, Balasubramaniam S, Zhang H, et al. FDA Approval Summary: Olaparib Monotherapy or in Combination with Bevacizumab for the Maintenance Treatment of Patients with Advanced Ovarian Cancer. Oncologist. 2021;26(1):e164-e172.

Binaschi M, Capranico G, De Isabella P, Mariani M, Supino R, Tinelli S, Zunino F. Comparison of DNA cleavage induced by etoposide and doxorubicin in two human small-cell lung cancer lines with different sensitivities to topoisomerase II inhibitors. Int J Cancer. 1990 Feb 15;45(2):347-52

Bryant HE, Schultz N, Thomas HD, Parker KM, Flower D, Lopez E, Kyle S, Meuth M, Curtin NJ, Helleday T. Specific killing of BRCA2-deficient tumours with inhibitors of poly(ADP-ribose) polymerase. Nature. 2005 Apr 14;434(7035):913-7.

Camarillo I G , Xiao F , Madhivanan S , et al. Low and high voltage electrochemotherapy for breast cancer: an in vitro model study. Electroporation-Based Therapies for Cancer, 2014:55-102.

Farmer H, McCabe N, Lord CJ, Tutt AN, Johnson DA, Richardson TB, Santarosa M, Dillon KJ, Hickson I, Knights C, Martin NM, Jackson SP, Smith GC, Ashworth A. Targeting the DNA repair defect in BRCA mutant cells as a therapeutic strategy. Nature. 2005 Apr 14;434(7035):917-21.

Hengel SR, Spies MA, Spies M. Small-Molecule Inhibitors Targeting DNA Repair and DNA Repair Deficiency in Research and Cancer Therapy. Cell Chem Biol. 2017 Sep 21;24(9):1101-1119.

Lin S, Choe J, Du P, Triboulet R, Gregory RI. The m(6)A Methyltransferase METTL3 Promotes Translation in Human Cancer Cells. Mol Cell. 2016 May 5;62(3):335-345.

Lok BH, Carley AC, Tchang B, Powell SN. RAD52 inactivation is synthetically lethal with deficiencies in BRCA1 and PALB2 in addition to BRCA2 through RAD51-mediated homologous recombination. Oncogene. 2013 Jul 25;32(30):3552-8.

Mendez-Dorantes C, Tsai LJ, Jahanshir E, Lopezcolorado FW, Stark JM. BLM has Contrary Effects on Repeat-Mediated Deletions, based on the Distance of DNA DSBs to a Repeat and Repeat Divergence. Cell Rep. 2020 Feb 4;30(5):1342-1357.e4.

Ouyang J, Yadav T, Zhang JM, Yang H, Rheinbay E, Guo H, Haber DA, Lan L, Zou L. RNA transcripts stimulate homologous recombination by forming DR-loops. Nature. 2021 Jun;594(7862):283-288. doi: 10.1038/s41586-021-03538-8.

Palleschi M, Tedaldi G, Sirico M, Virga A, Ulivi P, De Giorgi U. Moving beyond PARP Inhibition: Current State and Future Perspectives in Breast Cancer. Int J Mol Sci. 2021 Jul 23;22(15):7884

Tsai LJ, Lopezcolorado FW, Bhargava R, Mendez-Dorantes C, Jahanshir E, Stark JM. RNF8 has both KU-dependent and independent roles in chromosomal break repair. Nucleic Acids Res. 2020 Jun 19;48(11):6032-6052.

Welsh J E. Animal models for studying prevention and treatment of breast cancer. M.P. Conn (Ed.), Anim. Model. Stud. Prev. Treat. Breast Cancer, Academic Press (2013), pp. 997-1018.

Xia M, Guo Z, Hu Z. The Role of PARP Inhibitors in the Treatment of Prostate Cancer: Recent Advances in Clinical Trials. Biomolecules. 2021 May 12;11(5):722.

Yang F, Teves SS, Kemp CJ, Henik_off_ S. Doxorubicin, DNA torsion, and chromatin dynamics. Biochim Biophys Acta. 2014;1845(1):84-89. doi:10.1016/j.bbcan.2013.12.002.

Yang XH, Sladek TL, Liu X, Butler BR, Froelich CJ, Thor AD. Reconstitution of caspase 3 sensitizes MCF-7 breast cancer cells to doxorubicin- and etoposide-induced apoptosis. Cancer Res. 2001 Jan 1;61(1):348-54.